# LEARNING TO ATTRIBUTE WITH ATTENTION

## ABSTRACT

Given a sequence of tokens generated by a language model, we may want to identify the preceding tokens that *influence* the model to generate this sequence. Performing such *token attribution* is expensive; a common approach is to ablate preceding tokens and directly measure their effects. To reduce the cost of token attribution, we revisit attention weights as a heuristic for how a language model uses previous tokens. Naïve approaches to attribute model behavior with attention (e.g., averaging attention weights across attention heads to estimate a token's influence) have been found to be unreliable. To attain faithful attributions, we propose treating the attention weights of different attention heads as *features*. This way, we can *learn* how to effectively leverage attention weights for attribution (using signal from ablations). Our resulting method, Attribution with Attention (AT2), reliably performs on par with approaches that involve many ablations, while being significantly more efficient.

## 1 INTRODUCTION

When a language model generates content, it is guided by various prompts, provided context, and other preceding information. Pinpointing the specific tokens among these that *influence* the language model to generate a particular sequence can be valuable for understanding and debugging model behavior. Indeed, such attributions have been used to provide citations for generated statements (Cohen-Wang et al., 2024; Qi et al., 2024), identify biases (Vig et al., 2020), and assess the faithfulness of model-provided explanations (DeYoung et al., 2019; Lanham et al., 2023; Madsen et al., 2024).

When we say that a model is *influenced* by a particular set of tokens, we mean that *removing* these tokens would substantially affect its generation[1]. With this perspective, methods for (accurately) identifying influential tokens are often expensive. For example, the gradient of a model's output with respect to a preceding token approximates the token's influence but involves significant additional computation and memory to obtain (Yin & Neubig, 2022; Sarti et al., 2023). Ablating prior tokens is a way to directly measure their influence (Ribeiro et al., 2016a; Lundberg & Lee, 2017; Cohen-Wang et al., 2024) but requires an inference pass for each ablation; these ablations must be repeated for every example that we would like to attribute. Seeking to reduce the cost of attribution, we ask:

*Can we learn to (efficiently) predict the effect of ablating prior tokens on a model's generation?*

To answer this question, we consider the attention mechanism (Bahdanau et al., 2014), a key component of transformer language models (Vaswani et al., 2017). Examining attention weights is a widely used strategy for interpreting model behavior (Lee et al., 2017; Ding et al., 2017; Abnar & Zuidema, 2020; Vig et al., 2020). Intuitively, attention weights reflect how a language model utilizes information from preceding tokens when predicting the next one. However, prior work has illustrated that attention weights are often unreliable as explanations of model behavior (Jain & Wallace, 2019; Serrano & Smith, 2019; Vashishth et al., 2019). In particular, when attributing a model's generation to provided context, methods leveraging attention were shown to be accurate for attributing certain models but highly inaccurate for attributing others (Cohen-Wang et al., 2024).

To make effective use of attention weights for attribution, we are guided by the observation that different attention heads have different roles and utilities (Zheng et al., 2024). For example, Wu et al. (2024) identify "retrieval heads" responsible for copying information from context. Chuang et al. (2024) illustrate that certain heads are more useful than others for detecting hallucinations. These

---

[1]This follows from prior work on attribution (Lundberg & Lee, 2017; Ilyas et al., 2022) and measuring interpretability method faithfulness (Arras et al., 2017; DeYoung et al., 2019)

findings suggest that we may need to account for differences in roles and utilities when considering attention weights as a signal for attribution.

## 1.1 OUR CONTRIBUTIONS

We present an attribution method, Attribution with Attention (AT2), that treats the attention weights from different heads as *features* (Section 3). Specifically, we assign a (learnable) coefficient to each attention head signifying the extent to which we should rely on it to estimate influences. As a source of ground-truth for influence, we *ablate* random subsets of tokens and measure the corresponding effect on the model's generation. Learning attention head coefficients requires performing several ablations for each example in a training dataset; however, to then attribute any *new* generation, we can then just combine attention weights according to these coefficients.

We evaluate the effectiveness of AT2 in two settings: attributing a model's generation to provided context and to its intermediate thoughts (Section 4). We find that AT2 yields reliable attributions comparable to approaches that involve performing ablations for every example, while being several times more efficient. In particular, AT2 is effective even when naïve approaches for attributing with attention (e.g., averaging attention weights across layers and heads) are quite inaccurate.

## 2 PROBLEM STATEMENT

We begin by formalizing the task of *token attribution*: identifying the tokens that *influence* a model to generate a particular sequence (Section 2.1). Next, we present metrics for evaluating the quality of attribution methods (Section 2.2). Finally, we describe two examples of specific token attribution tasks that we will focus on: *context* and *thought attribution* (Section 2.3).

### 2.1 TOKEN ATTRIBUTION

**Setup.** Let $p_{\text{LM}}$ be an autoregressive language model, with $p_{\text{LM}}(t_i \mid t_1, \ldots, t_{i-1})$ denoting the probability that the next token is $t_i$ given preceding tokens $t_1, \ldots, t_{i-1}$. Suppose that we use $p_{\text{LM}}$ to generate a sequence of tokens $Y = y_1, \ldots, y_{|Y|}$ from an input sequence $X = x_1, \ldots, x_{|X|}$. Concretely, we generate the $i$'th token $y_i$ as follows:

$$y_i \sim p_{\text{LM}}(\cdot \mid x_1, \ldots, x_{|X|}, y_1, \ldots, y_{i-1}).$$

We let $p_{\text{LM}}(Y \mid X)$ denote the probability of generating the entire sequence $Y$ given $X$. The input sequence $X$ might be a user-provided prompt, a series of messages between a user and the language model itself, or a piece of writing for the language model to extend.

**Attributing generation to sources.** Token attribution involves understanding the extent to which different parts of the input sequence $X$ contribute to the generated sequence $Y$. We refer to these "parts of $X$" as *sources*, where each source is just a subsequence of $X$. The sources that we consider depend on the specific attribution task. They might be sentences, paragraphs, or individual tokens; they might span the entirety of $X$ or just a particular subset of interest. If, for example, the input sequence $X$ consists of a context and a query, we might be specifically interested in understanding how the model uses just the context.

We denote by $S = s_1, \ldots, s_{|S|}$ the set of sources associated with our attribution example, with each source $s_i \in \text{Subseq}(X)$. Our goal is to assign a score $\tau_i$ to each source $s_i$ that reflects the *influence* of $s_i$ when generating $Y$. We formalize this by defining a *token attribution method*:

**Definition 2.1** (Token attribution method). Suppose that we are given a language model $p_{\text{LM}}$, an input sequence $X$, a sequence $Y$ generated from $X$ by $p_{\text{LM}}$, and a set of sources $S$ over $X$. A *token attribution method* is a function $\tau(p_{\text{LM}}, X, S, Y) \in \mathbb{R}^{|S|}$ that assigns an attribution score to each source signifying its influence.

### 2.2 MEASURING THE QUALITY OF ATTRIBUTIONS

We have informally established that attribution scores should reflect the extent to which each source $s_i$ *influences* the model to generate the sequence $Y$. Following prior work (Lundberg & Lee, 2017; Arras et al., 2017; DeYoung et al., 2019; Ilyas et al., 2022), we adopt the perspective that if a source $s_i$ influences the model to generate $Y$, then *removing* $s_i$ should decrease the probability of generating $Y$. This perspective offers a natural methodology for evaluating the quality of attribution scores. Intuitively, when higher-scoring sources are removed, the probability of $Y$ should decrease more.

To operationalize this intuition, we first introduce the notion of a *source ablation*. Let $v \in \{0, 1\}^{|S|}$ be an *ablation vector* indicating a subset of the sources to ablate. We denote by $\text{ABLATE}(X, S, v)$ a

modified input sequence in which we remove every source $s_i$ for which $v_i = 0$. Our first metric, the *top-$k$ drop* (Cohen-Wang et al., 2024), measures the decrease in the log-probability of generating $Y$ when we ablate the $k$ highest-scoring sources:

**Definition 2.2** (*Top-$k$ drop*). Let $\tau$ be an attribution method. Let $v_{\text{top-}k}(\tau)$ be an ablation vector that excludes the $k$ highest-scoring sources according to $\tau$. Then the *top-$k$ drop* is the decrease in the log-probability of generating $Y$ when these top-$k$ sources are ablated:

$$\text{Top-}k\text{-drop}(\tau) := \underbrace{\log p_{\text{LM}}(Y \mid X)}_{\text{original log-probability}} - \underbrace{\log p_{\text{LM}}(Y \mid \text{ABLATE}(X, S, v_{\text{top-}k}(\tau)))}_{\text{log-probability with top-}k\text{ sources ablated}}.$$

The top-$k$ drop evaluates the ability of an attribution method to identify the $k$ most important sources. We would also like to consider whether the attribution scores of an *arbitrary* set of sources reflect the effect of ablating them. Specifically, we treat each attribution score as a prediction for the effect of ablating the corresponding source. We ablate *random* subsets of sources and measure the correlation between the actual and predicted effects of these ablations. This metric, proposed by Park et al. (2023), is known as the *linear datamodeling score* (see Section B.2 for details).

### 2.3 EXAMPLES OF TOKEN ATTRIBUTION TASKS

We now describe two examples of token attribution tasks that will be of interest to us. In Section 4, we will be evaluating the effectiveness of attribution methods on these tasks.

**Context attribution.** The goal of *context attribution* (Cohen-Wang et al., 2024) is to understand how a language model uses different pieces of information from a provided context when responding to a query. In this case, the input sequence $X$ consists of the provided context and the query. The generated sequence $Y$ is the response (or any span from it[2]). The sources might be the sentences, words, or tokens of the context. Sources with high attribution scores can be interpreted as "citations" for the model's response. These citations can help verify the correctness of the response, particularly when the context is long and difficult to read through entirely.

**Thought attribution.** As a second task, we propose *thought attribution*. Certain recent language models have been designed to generate "thoughts" before responding to a query (Jaech et al., 2024; Guo et al., 2025). These thoughts can significantly improve response quality. When models generate intermediate thoughts, the final response is often a highly distilled version of them. To understand how a model arrives at a particular conclusion presented in the response, it would thus be helpful to attribute this conclusion to its thoughts. In this setting, the input sequence $X$ consists of the original query and the model-generated thoughts. The generated sequence $Y$ to be attributed is the final response (or any span from it). The sources we consider are parts of the thoughts, e.g., sentences.

## 3 AT2: TOKEN ATTRIBUTION WITH ATTENTION

In this section, we describe Attribution with Attention (AT2), our method for token attribution. We first outline *attribution via surrogate modeling*, a general framework for attribution that has been applied extensively in prior work (Section 3.1). The key idea of this framework is to identify an easy-to-understand proxy that models the effects of ablating sources. By studying this proxy, we can attribute the model's generation to individual sources. Building on this framework, AT2 learns a surrogate model that uses attention weights as features, making attribution for unseen examples highly efficient (Section 3.2).

### 3.1 BACKGROUND: ATTRIBUTION VIA SURROGATE MODELING

To motivate AT2, we first describe *surrogate modeling* (Sacks et al., 1989), a technique widely used for attributing model behavior (Ribeiro et al., 2016a; Ilyas et al., 2022; Cohen-Wang et al., 2024). This technique aims to identify a simple proxy *surrogate model* that approximates a model's behavior while being easy to understand. This surrogate model can then be used to shed light on the model's behavior, and, in this case, to attribute behavior to individual sources.

To make this concrete, consider the token attribution setting: we would like to attribute a generation $Y$ to a set of sources $S$, each of which is a subsequence of the input $X$. Let $f(v)$ be the probability of generating $Y$ when ablating the sources according to $v$, that is,

$$f(v) := p_{\text{LM}}(Y \mid \text{ABLATE}(X, S, v)).$$

---

[2]To attribute a span, $Y$ would be this span and $X$ would include everything leading up to it.

> **Layer #19 Head #4**   **Layer #6 Head #22**   **Layer #25 Head #10**   **Generation of interest**
>
> **Context**: Hoping to increase their winning streak the Chargers moved to Reliant Stadium for an AFC duel with the Texans. In the first quarter the Chargers trailed early as RB Arian Foster got an 8-yard TD run. They soon replied with QB Philip Rivers making a 55-yard TD pass to WR Seyi Ajirotutu. They trailed again with kicker Neil Rackers nailed a 27-yard field goal, but took the lead after Rivers got an 11-yard TD pass to TE Randy McMichael. They fell behind again in the second quarter as Foster made a 2-yard TD run, followed by Rackers hitting a 21 and a 25-yard field goal. The Chargers eventually pulled themselves in front again, with Rivers finding McMichael again on a 12-yard TD pass. This was followed in the 4th quarter by Rivers' 28-yard TD pass to Ajirotutu (With a successful 2-point conversion as FB Mike Tolbert ran to the endzone). With the win, the Chargers went into their bye week at 4-5.
>
> **Query**: How many yards was Neil Rackers' shortest field goal?
>
> **Response** (from `Phi-3.5-mini`):
> Neil Rackers' shortest field goal in the game was 21 yards.
> He made three field goals in total, with distances of 27, 21, and 25 yards.

**Figure 1: Attention heads vary in their usefulness for attribution.** When visualizing attention weights of three individual heads, we observe that certain heads appear to be more useful for attribution than others. In particular, layer #19, head #4 assigns high attention weights to "27", "21" and "25" which are the field goal distances mentioned in the generation of interest. Meanwhile, layer #6, head #22 and layer #25, head #10 assign high attention weights to other, seemingly unrelated parts of the context. This example is from DROP (Dua et al., 2019) with a generation from `Phi-3.5-mini` (Abdin et al., 2024). Attention weights are averaged across the generation of interest and normalized by dividing by the maximum weight for each head.

We would like to understand how $f$ varies as a function of $v$; are there specific sources that signficantly affect $f$ when ablated? To answer this question, suppose that we could approximate $f$ with a *linear surrogate model* of the following form:
$$\hat{f}_w(v) \coloneqq \langle w, v \rangle.$$
To understand $f$, we could then simply examine the coefficients $w$ of $\hat{f}_w$. Each coefficient $w_i$ would signify the effect of ablating the $i$'th source on the probability of generating $Y$—this value could be directly interpreted as an attribution score. While the assumption of a *linear* model may seem strong, prior work has empirically shown that this modeling choice is often effective[3]. Indeed, this type of surrogate model has been successfully applied to attribute model behavior to training examples (Ilyas et al., 2022), features (Lundberg & Lee, 2017) and model internals (Shah et al., 2024).

**Learning a linear surrogate model.** With a linear surrogate model of the form $\hat{f}_w$, our remaining goal is to find parameters $w$ to approximate $f$ as well as possible. We do so by sampling random ablations and optimizing the parameters $w$ to minimize the empirical difference between $\hat{f}_w$ and $f$, i.e., optimizing $\hat{f}_w$ to predict the effects of ablations. Given an example $(p_{\text{LM}}, X, S, Y)$, this process consists of the following steps:

1. Sample $m$ ablation vectors $v^{(1)}, \ldots, v^{(m)} \in \{0, 1\}^{|S|}$.
2. For each ablation $v^{(j)}$, compute $f(v^{(j)})$, the corresponding probability of generating $Y$.
3. Find parameters $\hat{w}$ that minimize a loss $\mathcal{L}$ (e.g., MSE) between $f$ and $\hat{f}_w$:
$$\hat{w} = \arg\min_w \mathcal{L}(\{f(v^{(j)})\}_j, \{\hat{f}_w(v^{(j)})\}_j).$$

Before proceeding, we highlight a subtle yet important property of this type of surrogate model: it is *example-specific*. It approximates a model's behavior for a certain example and can only provide attribution scores for the particular sources associated with this example. As a result, to perform attribution for any given example, we would need to learn a new surrogate model from scratch (which involves performing several ablations to learn from).

### 3.2 LEARNING A GENERALIZABLE SURROGATE MODEL USING ATTENTION

In Section 3.1, we reviewed *surrogate modeling* as a method for attributing model behavior. Using a surrogate to model the effects of ablations is an effective approach across attribution settings. However,

---

[3]Specifically, a linear surrogate model can attain a high LDS (Definition B.1).

the standard instantiation of this approach—learning an *example-specific* surrogate model for each attribution—may be prohibitively costly. AT2's design is motivated by the following question:

*Can we learn to model the effects of ablations in a way that extends beyond a specific example?*

If so, we would just need to learn the surrogate model's parameters once. Attributing a *new* example would be cheap (assuming the surrogate model is cheap to evaluate).

**Learning a generalizable surrogate model.** Recall that the example-specific surrogate model requires learning a coefficient $w_i$ for each source, which is interpreted as an attribution score for this source. Our approach to identify a surrogate model is as follows: instead of directly learning attribution scores $w$ for particular sources, we learn a parameterized function $w_\theta(X, S, Y)$ that estimates scores *across* examples and sources (for a particular language model $p_{\text{LM}}$). This score-estimating function uses information about the language model $p_{\text{LM}}$, the input sequence $X$, the sources $S$, and the generated sequence $Y$ as its features. Its learnable parameters, $\theta$, are optimized to predict the effects of ablations (just as we optimize attribution scores $w$ directly for example-specific surrogate models). The resulting surrogate model for predicting the probability of generating $Y$ is

$$\hat{f}_\theta(v, X, S, Y) \coloneqq \langle w_\theta(X, S, Y), v \rangle.$$

In order to be useful, the score-estimating function $w_\theta$ needs to be both (1) cheap to compute and (2) effective at modeling the effects of ablations. We begin by designing features for $w_\theta$, that is, identifying information that is useful for attribution and cheap to extract from $p_{\text{LM}}$, $X$, $S$, and $Y$.

**Designing features for $w_\theta$.** To obtain features for $w_\theta$ that are both cheap to compute and provide signal for attribution, we consider using artifacts from the model's generative process (these would require *no* additional computation). In particular, for transformer models (Vaswani et al., 2017), attention weights are a natural candidate for the features of $w_\theta$. Prior work suggests that specific attention heads are responsible for particular behaviors (Zheng et al., 2024) and have distinct utilities (Chuang et al., 2024). We illustrate this in Figure 1: when we visualize the attention weights of a few attention heads, we observe that some appear to be more useful for attribution than others.

Motivated by this observation, we use the attention weights of individual heads as features for attribution. This way, we can learn to rely on specific heads that more accurately reflect the model's use of preceding tokens. To formalize this, we begin by introducing notation for describing attention weights. Recall that the transformer architecture consists of $L$ layers with $H$ heads in each layer. We write $\text{Attn}(X, Y, i, j) \in \mathbb{R}^{L \times H}$ to denote the attention weights (across layers and heads) assigned to the $i$-th token of $X$ when generating the $j$-th token of $Y$. We are interested in attributing $Y$ to individual sources $s_1, \ldots, s_{|S|} \in \text{Subseq}(X)$. Thus, we aggregate weights over $Y$ and a source $s$ as follows (overloading the "Attn" notation):

$$\text{Attn}(X, Y, s) \coloneqq \frac{1}{|Y|} \sum_{j=1}^{|Y|} \sum_{i \in s} \text{Attn}(X, Y, i, j).$$

We are now ready to express $w_\theta$ using these aggregated attention weights as features. The $i$-th element of $w_\theta$, i.e., the attribution score for the $i$-th source $s_i$ is given by

$$w_\theta(X, S, Y)_i = \sum_{\ell=1}^{L} \sum_{h=1}^{H} \theta_{\ell h} \text{Attn}(X, Y, s_i)_{\ell h}.$$

Here, $\theta \in \mathbb{R}^{L \times H}$ are coefficients specifying the extent to which we rely on each head.

**Learning the parameters $\theta$.** The next step is to learn parameters $\theta$ such that $w_\theta$ predicts effective attribution scores. To do so, we apply the same methodology as in Section 3.1: we perform a small number of random ablations and learn $\theta$ to predict the effects of these ablations (via the surrogate model $\hat{f}_\theta$). However, instead of learning from a single example, we learn from a *dataset* of examples (each consisting of an input sequence $X$, a set of sources $S$ and a generated sequence $Y$). In doing so, we hope to identify parameters $\theta$ that generalize to unseen examples. We summarize the resulting method, AT2, in Algorithm 1 (see Section B.6 for implementation details).

**Visualizing the attention weights of heads used and ignored by AT2.** Previously, in Figure 1, we observed that certain attention heads appear to be more useful for attribution than others. In Figure 4, we find that the attention heads that AT2 learns to rely on align with this intuition: the highest-coefficient head attends to tokens very relevant to the generation of interest, while the lowest-coefficient head attends to other, seemingly unrelated tokens.

---

**Algorithm 1** Attribution with Attention (AT2)

---

1: **Input:** Transformer language model $p_{\text{LM}}$, training dataset of $n$ examples $\{X^{(i)}, S^{(i)}, Y^{(i)}\}_{i=1}^n$, number of ablations $m$ to perform per example, loss function $\mathcal{L}$ measuring surrogate quality.
2: **Output:** Learned attribution method $\hat{\tau}_{\text{AT2}}$
3: $w_\theta(X, S, Y)_i := \sum_{\ell,h} \theta_{\ell h} \text{Attn}(X, Y, s_i)_{\ell h}$     $\triangleright$ Score estimator with attn. weights as features
4: $\hat{f}_\theta(v, X, S, Y) := \langle w_\theta(X, S, Y), v \rangle$     $\triangleright$ Surrogate model (linear in the ablation vector $v$)
5: **for** $i \in \{1, \ldots, n\}$ **do**
6:      **for** $j \in \{1, \ldots, m\}$ **do**
7:         Sample ablation $v^{(j)}$ from $\{0, 1\}^{|S^{(i)}|}$
8:      **end for**
9:      $f^{(i)}(v) := p_{\text{LM}}(Y^{(i)} \mid \text{ABLATE}(X^{(i)}, S^{(i)}, v))$     $\triangleright$ Probability of $Y^{(i)}$ when ablating by $v$
10:      $\hat{f}_\theta^{(i)}(v) := \hat{f}_\theta(v, X^{(i)}, S^{(i)}, Y^{(i)})$     $\triangleright$ Surrogate prediction when ablating by $v$
11:      $\mathcal{L}^{(i)}(\theta) := \mathcal{L}(\{\hat{f}_\theta^{(i)}(v^{(j)})\}_j, \{f(v^{(j)})\}_j)$     $\triangleright$ Loss for $i$'th example (across $m$ ablations)
12: **end for**
13: $\hat{\theta} \leftarrow \arg\min_\theta \frac{1}{n} \sum_{i=1}^n \mathcal{L}^{(i)}(\theta)$     $\triangleright$ Optimize $\theta$ to minimize loss across examples
14: $\hat{\tau}_{\text{AT2}}(p_{\text{LM}}, X, S, Y) := w_{\hat{\theta}}(X, S, Y)$     $\triangleright$ Treat learned score estimator as attribution method
15: **return** $\hat{\tau}_{\text{AT2}}$

---

## 4 EVALUATING TOKEN ATTRIBUTION METHODS

To evaluate the effectiveness of AT2, we consider two token attribution tasks: *context attribution* (Section 4.1) and *thought attribution* (Section 4.2). See Sections B.1 and B.3 to B.6 for additional details and Section A.5 for more fine-grained evaluations.

**Baselines.** In addition to AT2, we evaluate the following attribution methods:

1. *Example-specific surrogate modeling (ESM)*: We attribute using an example-specific linear surrogate model as described in Section 3.1. With a sufficient number of ablations, an example-specific surrogate model is an oracle of sorts for AT2 (which learns a surrogate model to generalize across examples). We follow Cohen-Wang et al. (2024) using $32, 64, 128$ and $256$ ablations.

2. *Average attention*: As a baseline for leveraging attention weights, we consider simply averaging the attention weights over attention heads (AT2 learns a coefficient for each head). This strategy has been used to explain model behavior (Kim et al., 2019; Sarti et al., 2023). In the context attribution setting of Cohen-Wang et al. (2024), average attention performed the best among attention-based methods and yielded effective attributions (i.e., close in performance to surrogate modeling) for some models but not others.

3. *Gradient $\ell_1$-norm*: The gradient with respect to a preceding token is a measure of its influence (Simonyan et al., 2013; Li et al., 2015; Smilkov et al., 2017). As another baseline, we compute the gradient of the log-probability of the generated sequence with respect to the embeddings of preceding tokens. To score each source, we sum the $\ell_1$-norm of the gradients for its tokens.

**Experiment setup.** For each of the context and thought attribution settings, we consider several datasets. For each dataset, we sample $400$ examples from the validation split for evaluation. Each example consists of a prompt to which we generate a response (and intermediate thoughts where applicable) using a language model fine-tuned to follow instructions. When the response consists of multiple sentences (according to an off-the-shelf sentence tokenizer (Bird et al., 2009)), we consider each sentence as a separate attribution target. We evaluate each method by measuring its top-5 log-probability drop (Definition 2.2) and LDS (Definition B.1), averaged across examples and targets.

**Training AT2.** A key ingredient of AT2 is the training dataset used to learn the attention head coefficients $\theta$. In some cases, we may have access to a dataset matching the distribution of examples we would like to attribute. In others, we may only have access to, e.g., a generic instruction following dataset. To evaluate AT2, we consider both settings. We write AT2 (task-specific) to denote training using examples from the task of interest, and AT2 (general) to denote training using examples from a generic dataset. We train AT2 using $2,000$ examples sampled from the training split of each dataset, performing $32$ ablations for each example. Hence, the cost of training AT2 in this way is roughly equivalent to the cost of $64,000$ forward passes of the language model. We always train AT2 using individual tokens as sources (we find that the resulting parameters transfer well to less fine-grained sources, e.g., sentences) and sentences as attribution targets.

**Figure 2: Qualitative comparison of token attributions.** We visualize the attribution scores (blue) of different methods for a particular generated statement (yellow) in a context attribution setting for `Phi-3.5-mini` (with individual tokens as sources). AT2 (trained on a generic dataset) and ESM (with 256 ablations) yield similar attributions with high scores for tokens related to the generated statement of interest, while average attention assigns the highest score to a seemingly arbitrary token. See Section A.4 for additional examples.

### 4.1 CONTEXT ATTRIBUTION SETUP

Context attribution (see Section 2.3) is the task of understanding how language models use different parts of a provided context when generating a response. In their work, Cohen-Wang et al. (2024) focus on *sentences* as sources; we consider both sentences and individual tokens from the context as sources. We attribute the variants of `Llama-3.1-8B` (Dubey et al., 2024) and `Phi-3.5-Mini` (Abdin et al., 2024) fine-tuned to follow instructions. For evaluation, we consider *CNN DailyMail* (Nallapati et al., 2016), a news article summarization dataset, *Hotpot QA* (Yang et al., 2018), a multi-hop question answering dataset, *Natural Questions* (Kwiatkowski et al., 2019), a long-context question answering dataset, and *MS MARCO* (Nguyen et al., 2016), a multi-document question answering dataset. As a generic context attribution dataset for training AT2, we consider *Dolly 15k* (Conover et al., 2023), an instruction following dataset (filtered to only examples with a context).

### 4.2 THOUGHT ATTRIBUTION SETUP

Thought attribution (see Section 2.3) is the task of understanding how language models use intermediate thoughts when generating a response. In this evaluation, we attribute to both intermediate thoughts and context (when present). As in the context attribution evaluation, we consider both sentences and individual tokens as sources. We attribute a distilled reasoning model, `DeepSeek-R1-Qwen-7B` (Guo et al., 2025). For evaluation, we consider *DROP* (Dua et al., 2019), a context-based question answering dataset focused on reasoning, and *Global Opinions QA* (Durmus et al., 2023), a dataset of opinion questions on global issues. As a generic dataset for training AT2, we consider *AGIEval* (Zhong et al., 2023), a dataset of reasoning problems spanning law, logic and math.

### 4.3 RESULTS

We provide a qualitative comparison of the attributions of different methods in Figure 2. In Figures 3a and 3b, we find that AT2 consistently outperforms the gradient baseline, which requires substantial additional computation, as well as the average attention baseline (a naïve application of attention for attribution). AT2 performs comparably to example-specific surrogate modeling with a substantial number of ablations, even when average attention performs quite poorly (as is the case for `Phi-3.5-mini`). We observe that the learned parameters $\theta$ (signifying reliance on each attention head) are fairly robust to the distribution of training examples: AT2 performs similarly when trained using examples from the task of interest (task-specific), or when trained using examples from a

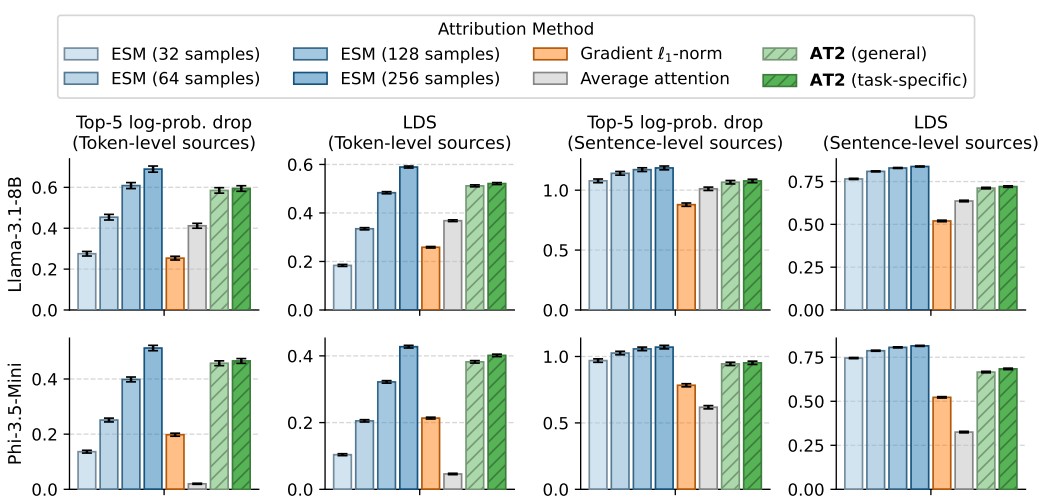

(a) **Evaluating context attributions.** We report the log-probability drop (Definition 2.2) and LDS (Definition B.1) for different attribution methods applied to `Llama-3.1-8B` and `Phi-3.5-Mini`. We consider *individual tokens* as sources (left) and *sentences* as sources (right). Metrics are averaged across different context attribution tasks (see Section 4.1).

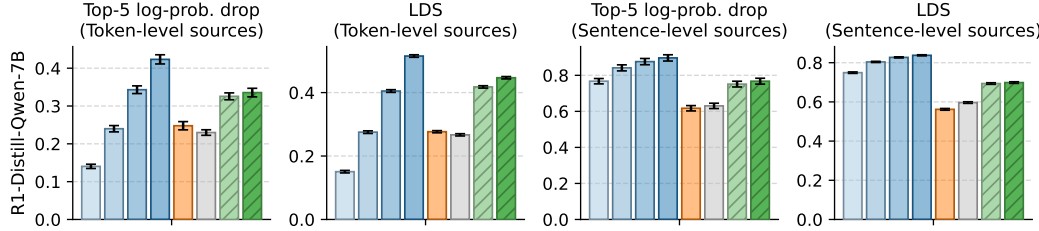

(b) **Evaluating thought attributions.** We report the log-probability drop (Definition 2.2) and LDS (Definition B.1) for different attribution methods applied to `DeepSeek-R1-Qwen-7B`. We consider *individual tokens* as sources (left) and *sentences* as sources (right). Metrics are averaged across different thought attribution tasks (see Section 4.2).

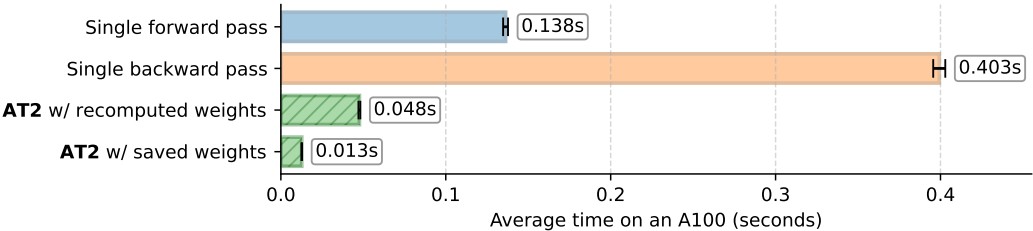

(c) **Comparing efficiency.** We report the average running time for a single forward pass, a single backward pass, and for AT2 for `Llama-3.1-8B` on Hotpot QA (on an A100 GPU). We consider AT2 with saved attention weights and with recomputed attention weights (for certain attention implementations, weights cannot be easily saved).

**Figure 3: Evaluating token attributions.** We report performance metrics for different attribution methods in context attribution (Figure 3a) and thought attribution (Figure 3b) settings. AT2 performs similarly when trained on examples from the task of interest (task-specific) or when trained on examples from a generic task (general). It consistently outperforms the gradient and average attention baselines and performs comparably to example-specific surrogate modeling (ESM) with a substantial number of ablations. In Figure 3c, we compare the running time of AT2 to a single forward pass (ESM uses $\geq 32$) and a single backward pass (which the gradient method requires). See Section A.5 for additional evaluations.

generic dataset (general). Furthermore, despite being trained only with individual tokens as sources, AT2 performs well when considering sentences as sources.

**Attribution efficiency.** In Figure 3c, we compare the running time of our implementation of AT2 to a single forward pass and a single backward pass on a single A100 GPU. We consider two settings for AT2: one in which we have access to saved attention weights and one in which we recompute the relevant attention weights from hidden states. The latter setting arises when using attention implementations that do not store the entire attention matrix (e.g., FlashAttention (Dao et al., 2022)). Even when recomputing attention weights, AT2 is more than $2\times$ faster than one forward pass and $8\times$ faster than one backward pass. This is substantially faster than methods involving ablations (which require $\geq 32$ forward passes for similar performance) or gradient-based methods.

**Downstream utility.** To assess the downstream utility of AT2, in Section A.3, we use it to prune unimportant pieces of context in a context-based question answering setting. Doing so improves answer quality across models on HotpotQA (Yang et al., 2018).

## 5 RELATED WORK

**Attributing generation to preceding tokens.** Attributing model generation to preceding tokens is valuable for a variety of downstream tasks: providing citations for generated statements (Cohen-Wang et al., 2024; Qi et al., 2024; Liu et al., 2024a), detecting hallucinations (Chuang et al., 2024), identifying biases (Vig et al., 2020), and assessing the faithfulness of model-provided explanations (Lanham et al., 2023; Madsen et al., 2024). Methods for attribution include performing ablations (DeYoung et al., 2019; Lanham et al., 2023), computing gradients (Yin & Neubig, 2022; Enguehard, 2023), examining attention weights (Vig et al., 2020), and using embedding similarities (Phukan et al., 2024). Following prior work on attribution and faithfulness in interpretability, we adopt the perspective that if a source is influential, then *removing* it should significantly affect the model's behavior (Lundberg & Lee, 2017; Ilyas et al., 2022; Arras et al., 2017; DeYoung et al., 2019; Madsen et al., 2021).

**Efficient attribution and explanation.** By learning attention head coefficients once and then performing attribution across examples, we are *amortizing* the cost of attribution. While the cost of training AT2 is high, the cost of attributing an unseen example is low. Prior work has similarly reduced the cost of explanation by training an explainer model (like our surrogate model) to mimic the behavior of an existing expensive explanation method (Schwarzenberg et al., 2021; Situ et al., 2021). For example, Jethani et al. (2021) approximate Shapley values (Shapley et al., 1953) in this manner, which normally require several ablations to approximate well. Covert et al. (2024) propose a general framework for amortizing the cost of attribution by learning from noisy attributions. While these methods use a separate learned neural network over the input to attribute or explain model behavior, we use just a linear model over attention weights. See Section C.1 for more related work.

**Using attention to explain model behavior.** Visualizing attention weights is a common strategy for interpreting model behavior (Lee et al., 2017; Ding et al., 2017; Abnar & Zuidema, 2020; Vig et al., 2020). However, prior work has cast doubt on the reliability of attention weights as explanations (Jain & Wallace, 2019; Wiegreffe & Pinter, 2019; Serrano & Smith, 2019). For example, Jain & Wallace (2019) find that attention weights are not well-correlated with other measures of importance such as gradients and can frequently be manipulated without substantially changing a model's output.

In this work, we are interested in whether attention weights across heads can be used to predict the effect of ablating a source. For transformer models, a typical approach is to average attention weights across heads (Kim et al., 2019; Sarti et al., 2023). In a context attribution setting (one of the token attribution settings we consider), this strategy has been found to be effective for attributing some models but yields very inaccurate attributions for other models (Cohen-Wang et al., 2024). In contrast, our observations suggest that the signal from attention weights *can* reliably attribute model behavior (provided that we account for the differences in behaviors of different attention heads).

## 6 CONCLUSION

We propose a token attribution method, Attribution with Attention (AT2), which models the effects of source ablations by treating the attention weights of different attention heads as features. By learning to combine these weights, AT2 attains comparable attribution quality to previous methods that require significantly more computation. AT2 can be applied to a variety of tasks, including context and thought attribution.

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

# Appendices

## A    ADDITIONAL RESULTS

### A.1    VISUALIZING THE ATTENTION WEIGHTS OF HEADS USED AND IGNORED BY AT2.

In Figure 4, we visualize the attention weights for attention heads that AT2 learns to rely on (high learned coefficient) and ignore (low-magnitude learned coefficient).

---

Highest-coefficient head       Lowest-coefficient head       Generation of interest

**Context**: Hoping to increase their winning streak the Chargers moved to Reliant Stadium for an AFC duel with the Texans. In the first quarter the Chargers trailed early as RB Arian Foster got an 8-yard TD run. They soon replied with QB Philip Rivers making a 55-yard TD pass to WR Seyi Ajirotutu. They trailed again with kicker Neil Rackers nailed a 27-yard field goal, but took the lead after Rivers got an 11-yard TD pass to TE Randy McMichael. They fell behind again in the second quarter as Foster made a 2-yard TD run, followed by Rackers hitting a 21 and a 25-yard field goal. The Chargers eventually pulled themselves in front again, with Rivers finding McMichael again on a 12-yard TD pass. This was followed in the 4th quarter by Rivers' 28-yard TD pass to Ajirotutu (With a successful 2-point conversion as FB Mike Tolbert ran to the endzone). With the win, the Chargers went into their bye week at 4-5.

**Query**: How many yards was Neil Rackers' shortest field goal?

**Response** (from `Phi-3.5-mini`):
Neil Rackers' shortest field goal in the game was 21 yards.

He made three field goals in total, with distances of 27, 21, and 25 yards.

---

**Figure 4: AT2 identifies attention heads that appear useful for attribution.** The attention head with the highest coefficient (as learned by AT2) attends to tokens that seem very relevant to the generation of interest. On the other hand, the head with the lowest-magnitude coefficient attends to other, seemingly unrelated parts of the context. To learn these coefficients, AT2 is trained on *Dolly 15k* (Conover et al., 2023) (see Section 4 for details). The details are otherwise identical to Figure 1. See Section A.2 for the coefficients themselves.

### A.2    VISUALIZATION OF LEARNED COEFFICIENTS

In Figure 5, we visualize the learned coefficients of different models trained on their corresponding generic datasets. We do not observe a consistent pattern for high-magnitude coefficients—for `Llama-3.1-8B` and `DeepSeek-R1-Qwen-7B`, the high-magnitude coefficients are more uniformly distributed across layers, while for `Phi-3.5-mini`, the high-magnitude coefficients are well-concentrated in the middle layers.

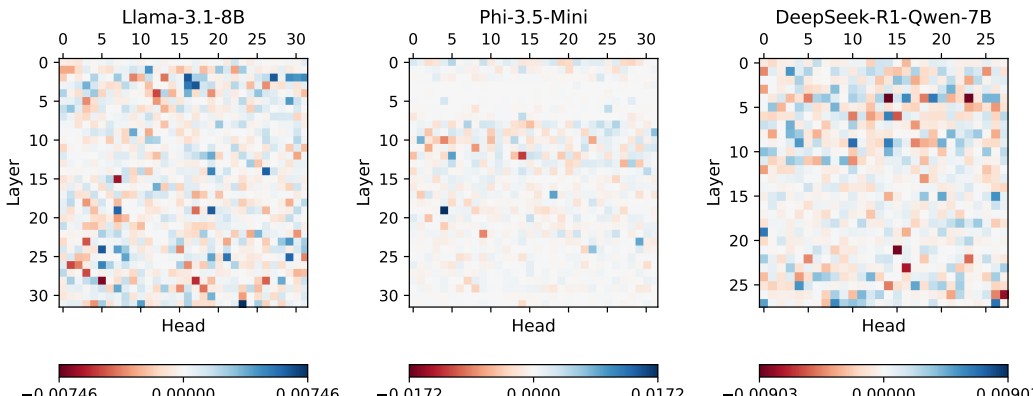

**Figure 5: Visualizing learned attention head coefficients.** We visualize the learned coefficients of AT2 for different models trained on their corresponding generic datasets (Dolly 15K for `Llama-3.1-8B` and `Phi-3.5-mini`, and AGIEval for `DeepSeek-R1-Qwen-7B`). While the coefficients for `Phi-3.5-mini` seems to have the highest magnitude for middle layers, the high-magnitude coefficients for `Llama-3.1-8B` and `DeepSeek-R1-Qwen-7B` are more uniformly distributed.

### A.3 APPLYING AT2 TO IMPROVE RESPONSE QUALITY

Beyond directly measuring the attribution performance of AT2, we examine its utility in a downstream application of context attribution: pruning the context to improve response quality. Specifically, Cohen-Wang et al. (2024) find that in a context-based question answering setting, pruning away parts of the context that are unused by the model (according to their attribution method) can improve performance. This aligns with previous observations that language models often struggle to answer questions when the relevant information is within long contexts (Peysakhovich & Lerer, 2023; Liu et al., 2024b). The attribution method proposed by Cohen-Wang et al. (2024), an example-specific surrogate modeling method, requires several inference passes (at least 32) to identify the most important parts of the context. Hence, improving performance comes at a substantially increased inference cost.

In Figure 6, we investigate whether attributions from AT2 can be used to prune the context to improve response quality on Hotpot QA. Hotpot QA is a multi-hop question answering dataset in which the answer involves information from two different passages among several provided passages (most of which are "distractors" that are irrelevant to the question). In this setting, if we provide the model with just the two relevant passages ("Oracle" in Figure 6), it performs substantially better than when provided with the entire context ("Baseline" in Figure 6). We find that pruning according to AT2 attributions improves response quality and is more effective than using attributions from example-specific surrogate modeling with 32 ablations and average attention. When using AT2, the cost of these performance gains (besides re-generating the answer) is substantially less than a single additional forward pass. We provide additional details in Section B.7.

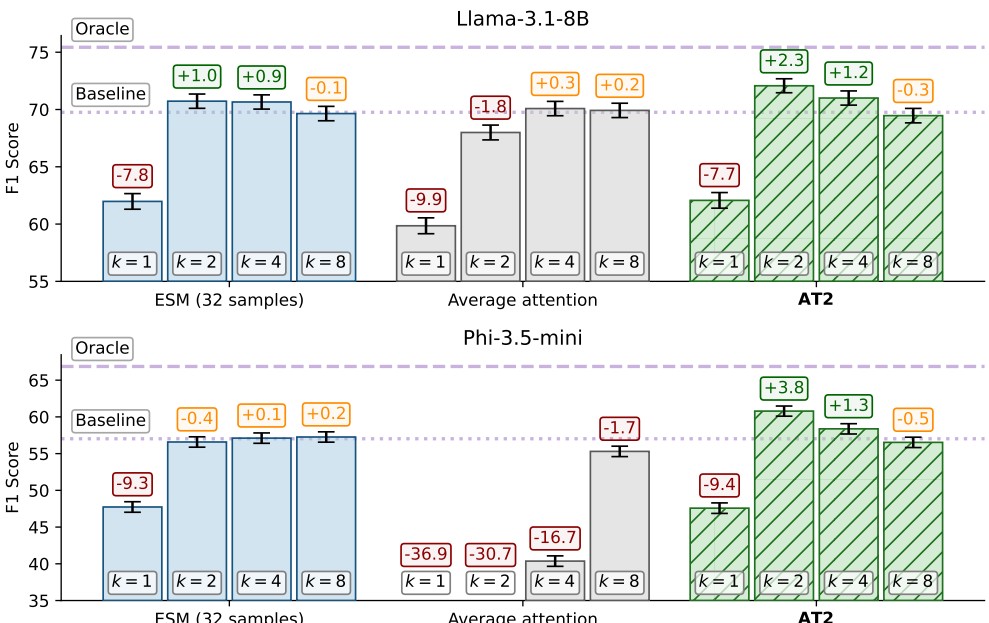

**Figure 6: Improving response quality by pruning the context.** Using attributions from AT2 to prune away less important parts of the context improves response quality on HotpotQA. We compare AT2 to example-specific surrogate modeling (ESM) with 32 ablations and average attention for different numbers of retained passages $k$, finding that AT2 improves performance more while being less costly than a single inference pass. "Baseline" denotes the performance of the model without context pruning, while "Oracle" denotes the performance of the model when provided only with the ground-truth sources. The F1 score is averaged over $4,000$ examples from the HotpotQA validation set.

## A.4 EXAMPLES OF ATTRIBUTIONS

To provide a more extensive qualitative evaluation of different attribution methods, we visualize the attribution scores of random examples from different datasets for different models and different sources types. To visualize attributions, we highlight sources according to their attribution scores (normalizing by the maximum attribution score for each example). We only show sources for which the attribution score is above $0.2$ of the maximum attribution score, or also surrounding sources in the case of token-level sources. We show context attribution examples in Figures 7 to 14 and thought attribution examples in Figures 15 to 18.

**AT2 (general)**

**Context**: ... of the week. The White House fired back hours later ... and lesbians . White House Press Secretary Josh Earn ... ve seen the governor and other Indiana officials in damage-control mode here.' ... they want to discriminate against gays and les ... week - have voiced concern over the effect of the law, and some states have barred government-funded travel to Indiana. NASC ... Arkansas. At the White House, Earnest ... surprise that 'we see business leaders saying that they are rel ... be subjected to greater discrimination just because of ... payer-funded travel to Indiana this week in ... on Monday to introduce a travel ban and was followed by Washington ... in a statement. Connecticut's governor, Dannel ... organization would back a clarification in the law explaining that ...' to the state and potentially to its economic future. ...

**Average attention**

**Context**: ... on his desk by the end of the week. ... has been meeting with lawmakers 'around the clock ... legislation flies in the face of the kinds of ... being served? Gay weddings are ground-zero ... back a clarification in the law explaining that it ' ... that 'it certainly wasn't my intent' L ... critics and many in the national media' for getting ...

**ESM (256 samples)**

**Context**: ... end of the week. The White House fired back hours later ...,' he told reporters during a press conference, ... and lesbians . White House Press Secretary Josh E ... kinds of values that people all across the country strongly support ... ability to exercise their religious beliefs without shelving ... But the central controversy surrounding the law concerns whether or ... effect of the law, and some states have barred government-funded travel to Indiana. NASC ... in Arkansas. At the White House, Earn ... or even their employees could be subjected to greater discrimination just because of who ... payer-funded travel to Indiana this week in response ... on Monday to introduce a travel ban and was followed by ... and safety.' 'New York State has been, and ... BGT family members, friends and colleagues to ensure ... get the picture from who was around him when he signed ... marriage and promotes religious liberty, came to Pence ... day, saying that Indiana has been the target of ' ... participate in a ceremony that violates religious beliefs is completely un-American and uncivil,' ... the government to participate in a same-sex ceremony.' Perkins said his organization would back ... non-religious goods or services.' Best intent ... into motion could swing in the opposite direction. Indianapolis ...

**Query**: Summarize the article in up to three sentences.

**Response** (from `Phi-3.5-Mini`):
Indiana Governor Mike Pence seeks to clarify that a new religious-freedom law does not permit discrimination, amid criticism and protests claiming it allows businesses to deny services to gays and lesbians. The White House and other businesses express concerns over the law's potential to enable discrimination, leading to travel bans and calls for amendments. Pence defends the law, stating it was not intended to create a license to discriminate, while opponents argue it has caused significant harm to Indiana's reputation and economy.

**Figure 7:** Visualized attributions for a random example from CNN DailyMail for `Phi-3.5-Mini` with tokens as sources.

**AT2 (general)**

**Context**: ... online gambling firm linked to Harry Redknapp and two Premier ... online gambling firm, which signed Harry Redknapp to star in its adverts (pictured above) has ... fraud . The online firm announced deals with West Bromwich Albion and Leicester City last summer. It also signed former Q ... app to appear in its adverts, alongside Lock, Stock .... Last week, the UK gambling regulator The Gambling Commission suspended the firm's licence. The suspension led to Football League One ... firm 666Bet announced deals with West Brom and Leicester City last summer. It ... be.' Although the Gambling Commission refused to comment, it previously said the suspension of the Metro Play Limited ...

**Average attention**

**Context**: ... firm linked to Harry Redknapp and two Premier League football clubs has been arrested over a ... online gambling firm, which signed Harry Redknapp to star in its adverts (pictured above) has been ... fraud . The online firm announced deals with West Bromwich Albion and Leicester City last summer. It also signed former Q ... app to appear in its adverts, alongside Lock, Stock and .... Last week, the UK gambling regulator The Gambling Commission suspended the firm's licence. The suspension led to Football League One ... makers in August last year, terminating its deal for shirt sponsorship. It is not yet known whether the two Premier League clubs will follow suit. Speaking ... firm 666Bet announced deals with West Brom and Leicester City last summer. It also ... will be.' Although the Gambling Commission refused to comment, it previously said the suspension of the Metro Play Limited ... accepting comments on this article.

**ESM (256 samples)**

**Context**: ... online gambling firm linked to Harry Redknapp and two Premier .... Six other people were arrested, 13 properties were ... an online gambling firm, which signed Harry Redknapp to star in its adverts (pictured above) has ... arrested over a PS21m fraud . The online firm announced deals with West Bromwich Albion and Leicester City last summer. It ... regulator The Gambling Commission suspended the firm's licence. The ... firm 666Bet announced deals with West Brom and Leicester City last ... just did an advert for them. I don't know ... official Twitter account said the site was under maintenance due to ... seen circumstances' In another tweet, it says: ' ... have the site and our licence back ASAP but cannot comment ... of the Metro Play Limited licence - which covers 666 ...

**Query**: Summarize the article in up to three sentences.

**Response** (from `Llama-3.1-8B`):
A director of online gambling firm 666Bet, Paul Bell, has been arrested as part of a PS21m fraud probe by HM Revenue & Customs and the National Crime Agency. The firm, which signed Harry Redknapp to star in its adverts and had deals with West Bromwich Albion and Leicester City, had its licence suspended by the UK Gambling Commission. The suspension has led to one of the firm's shirt sponsorship deals being terminated, and it is unclear whether the two Premier League clubs will follow suit.

**Figure 8:** Visualized attributions for a random example from CNN DailyMail for `Llama-3.1-8B` with tokens as sources.

---

**AT2 (general)**

**Context**: … This time she seems to have decided to fully embrace her womanhood as an asset in her quest for the White House and to trust that the voters will do the same. … This time around, there seems to be a shared assumption that, of course, a woman would make an excellent president, in part simply by virtue of being a woman. … If Hillary seems more comfortable running as a woman, it's partly because society at large seems more comfortable with a populist-tinged feminism than it was in 2008. … Women have been steadily making strides in the years since Hillary's 2008 campaign, and as they did, they smoothed the way for one of their own to run for the highest office in the land without having to play down her gender. Hillary is now wisely embracing her gender as a way of capturing the same "hope and change" historical quality of Obama's presidency. …

**Average attention**

**Context**: … Of course, Hillary hasn't officially announced that she will be running for president – and Universal Studios has not officially announced that there will be a sequel to the blockbuster "50 Shades of Grey." … Hillary's stressing of the importance of women's workplace issues both to the family and to the struggles of the middle-class puts the Republicans on the defensive as they try to co-opt middle-class economic issues. … Women have been steadily making strides in the years since Hillary's 2008 campaign, and as they did, they smoothed the way for one of their own to run for the highest office in the land without having to play down her gender. …

**ESM (256 samples)**

**Context**: … In 2008, she seemed to think that she had to reassure voters that a woman could be president, primarily by not accentuating the fact that she was a woman. This time around, there seems to be a shared assumption that, of course, a woman would make an excellent president, in part simply by virtue of being a woman. … Fortunately, the dynamics of the 21st century economy and the place of the family within it lend themselves to a family-friendly feminism. … If Hillary seems more comfortable running as a woman, it's partly because society at large seems more comfortable with a populist-tinged feminism than it was in 2008. … Women have been steadily making strides in the years since Hillary's 2008 campaign, and as they did, they smoothed the way for one of their own to run for the highest office in the land without having to play down her gender. …

**Query**: Summarize the article in up to three sentences.

**Response** (from `Phi-3.5-Mini`):
Hillary Clinton is embracing her womanhood as a key aspect of her 2016 presidential campaign, highlighting women's economic issues and the importance of family-friendly policies. She is set to speak at numerous events focused on women's issues during Women's History Month, marking a stark contrast to her 2008 campaign. Clinton's approach reflects a societal shift towards a more inclusive feminism, with her gender now seen as an asset rather than a point of contention.

**Figure 9:** Visualized attributions for a random example from CNN DailyMail for `Phi-3.5-Mini` with sentences as sources.

**AT2 (general)**

**Context**: A have-a-go-hero was so badly beaten and covered in blood after he challenged a car thief that his family thought he had been shot. Danny Mason, 37, was repeatedly punched in the face and left with a fractured cheek, broken nose and cut lip after he approached the would-be-thief in Blackburn, Lancashire. He had just left a family party at around 11.45pm on Saturday when he tried to stop the would-be-thief from stealing a number of cars. The 37-year-old, who now lives in Kidderminster, Worcestershire, was found slumped unconscious on the ground by family members and is now in so much pain that he cannot sleep or work. Danny Mason, 37, was repeatedly punched in the face and left with a fractured cheek, broken nose and cut lip after challenging a car thief . … He had just left a family party at around 11.45pm on Saturday when he approached the would-be thief and was viciously attacked in Parkes Way, Blackburn . …

**Average attention**

**Context**: A have-a-go-hero was so badly beaten and covered in blood after he challenged a car thief that his family thought he had been shot. Danny Mason, 37, was repeatedly punched in the face and left with a fractured cheek, broken nose and cut lip after he approached the would-be-thief in Blackburn, Lancashire. He had just left a family party at around 11.45pm on Saturday when he tried to stop the would-be-thief from stealing a number of cars. The 37-year-old, who now lives in Kidderminster, Worcestershire, was found slumped unconscious on the ground by family members and is now in so much pain that he cannot sleep or work. Danny Mason, 37, was repeatedly punched in the face and left with a fractured cheek, broken nose and cut lip after challenging a car thief . … 'Looking back, I thought I was doing the right thing by speaking to this man, but now I am in so much pain that I cannot sleep or work.' … Police have now released CCTV images of a white man around 6ft tall who they would like to speak to in connection with the incident. Mr Mason's sister Sue Bannister said that the family have been 'deeply shocked and upset' by the attack and asked anyone with information to come forward. … 'Danny had gone outside for a cigarette when he spotted a man who looked like he was trying to break into vehicles. … The 37-year-old (before the attack, left) was found slumped unconscious on the ground by family members and is now in so much pain that he cannot sleep or work (after the attack, right) Police described the attacker as white, between 6ft and 6ft 2in tall, of an average to medium build with short light brown or dark blonde hair . … He had just left a family party at around 11.45pm on Saturday when he approached the would-be thief and was viciously attacked in Parkes Way, Blackburn . … Anyone with information about the incident which happened at 11.45pm on Saturday should contact police on 101 quoting log number 1567 of February 28 or Crimestoppers on 0800 555 111.

**ESM (256 samples)**

**Context**: A have-a-go-hero was so badly beaten and covered in blood after he challenged a car thief that his family thought he had been shot. Danny Mason, 37, was repeatedly punched in the face and left with a fractured cheek, broken nose and cut lip after he approached the would-be-thief in Blackburn, Lancashire. He had just left a family party at around 11.45pm on Saturday when he tried to stop the would-be-thief from stealing a number of cars. … Danny Mason, 37, was repeatedly punched in the face and left with a fractured cheek, broken nose and cut lip after challenging a car thief . … 'Danny had gone outside for a cigarette when he spotted a man who looked like he was trying to break into vehicles. …

**Query**: Summarize the article in up to three sentences.

**Response** (from `Llama-3.1-8B`):
A 37-year-old man, Danny Mason, was brutally attacked by a car thief in Blackburn, Lancashire, after he tried to stop the individual from stealing vehicles. Mason suffered a fractured cheek, broken nose, and cut lip in the assault, which left him in so much pain that he cannot sleep or work. Police have released CCTV images of a suspect and are appealing for anyone with information to come forward to help identify the attacker.

**Figure 10:** Visualized attributions for a random example from CNN DailyMail for `Llama-3.1-8B` with sentences as sources.

AT2 (general)

**Context**: ... 16.Right On! was an American te ...

Average attention

**Context**: ... yle and culture published in the United States. It ... 1926 during the Harlem Renaissance. ... anzine three times (2008-2 ... 1963 into the early 197 ... the National Geographic Society in the United States. The ... 009/January 201 ...

ESM (256 samples)

**Context**: ... was a monthly print magazine dedicated to news, commentary, ... circulated English-language monthly magazine for the Asian ... ants and communities of color. The magazine ended print issue in ... !! was an African-American literary magazine published in ... 926 during the Harlem Renaissance. The ... Neale Hurston, Aaron Douglas, John P ... Bennett, Lewis Grandison Alexander, Countee ... quarters burned down, and the magazine ended. ... Fanzine three times (2008-2010). The magazine ended ... The magazine was published on a biweekly basis ... 16. Fredrikke Morck took ... 916. The magazine ended its publication in 1927 ... 2 and 1975 by Calvin Thomas ... Gothic Castle Publishing Company, distributed by Kable News ... Ken Beale edited the first three issues. W ... 1963 into the early 19 ... Although promoted and sold as a "monster magazine," ... and fanzines. Its advertising pages sold ... issue was published in Spring 1999. Regular publication of the magazine ended in December 2009, and the name was reused for a biannual newsstand publication. ... S. "Time" magazine. It was ... London. The magazine ended its run by eventually being am ... -form feature stories, photography, and illustrations ... itchfork's online content. The magazine ... l issues In November 2016 ... an American teen magazine first published by the Laufer Company in 1972 with editor/creator Jud ... 2011 and focused on African- ... railroads and short line railroads, and ... and modeling. The magazine ended publication in August 2 ...

**Query**: The publication of which magazine ended first, Right On! or Castle of Frankenstein?

**Response** (from `Phi-3.5-Mini`):
The publication of Right On! ended first. Right On! was a teen magazine that continued publishing until around 2011. On the other hand, Castle of Frankenstein, an American horror, science fiction, and fantasy film magazine, ended its publication in 1975. Therefore, Right On! ended its publication before Castle of Frankenstein.

**Figure 11:** Visualized attributions for a random example from Hotpot QA for `Phi-3.5-Mini` with tokens as sources.

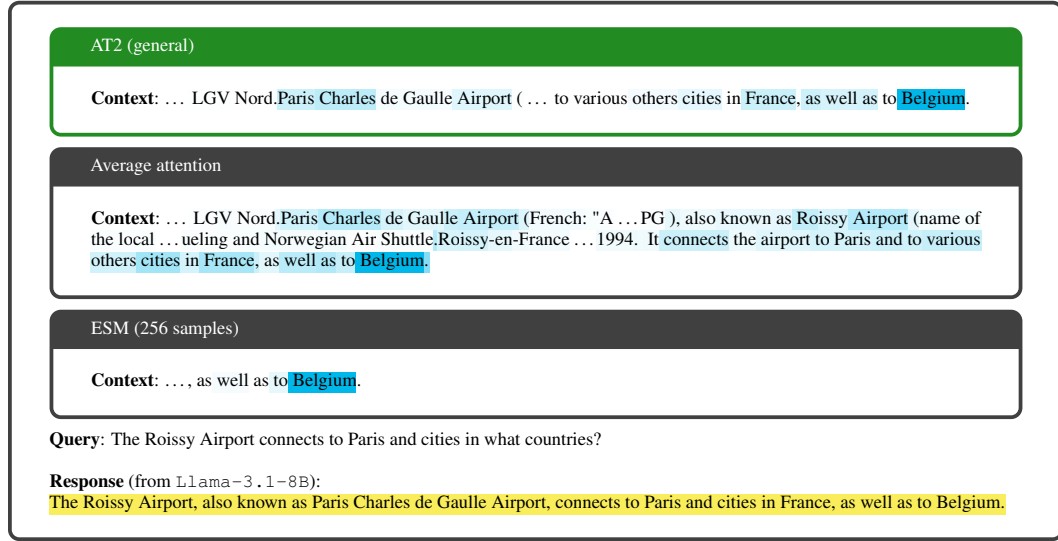

AT2 (general)

**Context**: ... LGV Nord.Paris Charles de Gaulle Airport ( ... to various others cities in France, as well as to Belgium.

Average attention

**Context**: ... LGV Nord.Paris Charles de Gaulle Airport (French: "A ... PG ), also known as Roissy Airport (name of the local ... ueling and Norwegian Air Shuttle.Roissy-en-France ... 1994. It connects the airport to Paris and to various others cities in France, as well as to Belgium.

ESM (256 samples)

**Context**: ..., as well as to Belgium.

**Query**: The Roissy Airport connects to Paris and cities in what countries?

**Response** (from `Llama-3.1-8B`):
The Roissy Airport, also known as Paris Charles de Gaulle Airport, connects to Paris and cities in France, as well as to Belgium.

**Figure 12:** Visualized attributions for a random example from Hotpot QA for `Llama-3.1-8B` with tokens as sources.

**AT2 (general)**

**Context**: Manga Life (mangaraihu , Manga Raifu ) is a manga magazine published monthly by Takeshobo in Japan since the November 1984 issue (published in October 1984). … Its original title was Gag da (giyaguda , Gyagu da ) , and the change to "Manga Life" was made to better compete with "Manga Time", a rival magazine published by Houbunsha. … Most of the series appearing in the magazine use the yonkoma format. … The magazine is released monthly on the 17th, though it sometimes appears on shelves slightly before or after that, depending on speed of actual distribution. … Dengeki G's Comic (Dian Ji Ji Ji G'skomitsuku , Dengeki Jizu Komikku ) is a Japanese seinen manga magazine published by ASCII Media Works. … The magazine was first published digitally on August 9, 2012 with volume 0, and started monthly publication with the following issue released on October 15, 2012. … Life was an American magazine that ran weekly from 1883 to 1936 as a humor magazine with limited circulation. " Time" owner Henry Luce bought the magazine in 1936, solely so that he could acquire the rights to its name, and launched a major weekly news magazine with a strong emphasis on photojournalism. … "Life" was published weekly until 1972, as an intermittent "special" until 1978, and as a monthly from 1978 to 2000. … Founded in 1993, the magazine is published eight times annually and covers the events, people, history and places of Newport County. … Newport Life Magazine is located at 101 Malbone Road in the Newport Daily News building. … Sylph (shiruhu , Shirufu ) is a Japanese shojo manga magazine published by ASCII Media Works (formerly MediaWorks) and is sold monthly. … The magazine was originally published on December 9, 2006 as a special edition version of MediaWorks' now-defunct "Dengeki Comic Gao! " under the title "Comic Sylph" (komitsukushiruhu , Komikku Shirufu , normally written as "comic SYLPH") as a quarterly publication. … On March 21, 2008, with the release of the sixth volume, the magazine was transferred over as a special edition version of ASCII Media Works' shonen manga magazine "Dengeki Daioh". … On May 22, 2008, the magazine became independent of "Dengeki Daioh" and was published as volume one of "Sylph" as the July 2008 issue as a bimonthly publication. … "Sylph" is one of the few magazines originally published by MediaWorks not under the "Dengeki" naming line, such as with "Dengeki Daioh", and "Dengeki G's Magazine", the first of which being "Active Japan" in 1995 which has been discontinued since 1998. … The magazine include articles about the following: history of slot machine, the principle of slot machine work, new strategies for playing video poker, casino reviews and latest casino news etc. … The Leeds Guide was a monthly "What's on" magazine published in Leeds, West Yorkshire, England from 1997 until 2012. … Originally a monthly A5 magazine printed in black and white, "The Leeds Guide" changed to an A4 fortnightly in 2003 and subsequently went back to being published monthly. … Sekai (Japanese: Shi Jie  "World") is a Japanese monthly political magazine published by Iwanami Shoten, which was founded in December 1945. … The magazine's founding principles were "peace and social justice, freedom and equality, and harmony and solidarity with the peoples of East Asia." … Great Lakes Life Magazine was a regional magazine that was published in Westfield, New York. … It was founded by editor-in-chief and publisher Rena Tran and began publication in February 2008 under the title Erie Life Magazine, but went out of production in 2011. …

**Average attention**

**Context**: Manga Life (mangaraihu , Manga Raifu ) is a manga magazine published monthly by Takeshobo in Japan since the November 1984 issue (published in October 1984). … Most of the series appearing in the magazine use the yonkoma format. … The magazine was first published digitally on August 9, 2012 with volume 0, and started monthly publication with the following issue released on October 15, 2012. … Newport Life Magazine is located at 101 Malbone Road in the Newport Daily News building. … On May 22, 2008, the magazine became independent of "Dengeki Daioh" and was published as volume one of "Sylph" as the July 2008 issue as a bimonthly publication. … "Sylph" is one of the few magazines originally published by MediaWorks not under the "Dengeki" naming line, such as with "Dengeki Daioh", and "Dengeki G's Magazine", the first of which being "Active Japan" in 1995 which has been discontinued since 1998. …

**ESM (256 samples)**

**Context**: … From April 2013 to April 2014, the magazine was released biweekly. … Life was an American magazine that ran weekly from 1883 to 1936 as a humor magazine with limited circulation. " … Sylph (shiruhu , Shirufu ) is a Japanese shojo manga magazine published by ASCII Media Works (formerly MediaWorks) and is sold monthly. … under the title "Comic Sylph" (komitsukushiruhu , Komikku Shirufu , normally written as "comic SYLPH") as a quarterly publication. … "Sylph" is one of the few magazines originally published by MediaWorks not under the "Dengeki" naming line, such as with "Dengeki Daioh", and "Dengeki G's Magazine", the first of which being "Active Japan" in 1995 which has been discontinued since 1998. … Strictly Slots is a monthly magazine aimed at slot machine and video poker players. … The magazine was founded in 1998. … Strictly Slots is published by the Casino Player Publishing, which also publishes "Casino Player" magazine. …

**Query**: Were both Life magazine and Strictly Slots magazine published monthly in 1998?

**Response** (from `Phi-3.5-Mini`):
No, they were not. Life magazine, which ran from 1883 to 1936 and then again from 1978 to 2000, was not published monthly in 1998. It was a weekly magazine during its second run and then became an intermittent "special" until 1978, after which it was published monthly.On the other hand, Strictly Slots is a monthly magazine that was founded in 1998. It is aimed at slot machine and video poker players and includes articles about the history of slot machines, strategies for playing video poker, casino reviews, and the latest casino news.So, only Strictly Slots was published monthly in 1998.

**Figure 13:** Visualized attributions for a random example from Hotpot QA for `Phi-3.5-Mini` with sentences as sources.

**AT2 (general)**

**Context**: … It is the first film adaptation of Woon Swee Oan's novel series "Si Da Ming Bu" (Si Da Ming Bu ; "The Four Great Constables"), which has previously been adapted to a television series. …

**Average attention**

**Context**: … It is the first film adaptation of Woon Swee Oan's novel series "Si Da Ming Bu" (Si Da Ming Bu ; "The Four Great Constables"), which has previously been adapted to a television series. … The Four III (Chinese: Si Da Ming Bu 3) is a Chinese-Hong Kong 3D wuxia film directed by Gordon Chan and Janet Chun. … It is the final installment of the trilogy based on Woon Swee Oan's novel series, after "The Four" (2012) and "The Four II" (2013).

**ESM (256 samples)**

**Context**: … It is the first film adaptation of Woon Swee Oan's novel series "Si Da Ming Bu" (Si Da Ming Bu ; "The Four Great Constables"), which has previously been adapted to a television series. …

**Query**: What is the English translation of the name of the novel series which the Four III is the final installment of?

**Response** (from `Llama-3.1-8B`):
The novel series is "Si Da Ming Bu" (Si Da Ming Bu ; "The Four Great Constables").

**Figure 14:** Visualized attributions for a random example from Hotpot QA for `Llama-3.1-8B` with sentences as sources.

**Query**: Thinking about your country's future, how important is it that: e. Copts and other religious minorities can practice their religion freely? Is it very important, somewhat important, not too important or not important at all?

**AT2 (general)**

**Thoughts**: … the nation's unity or harmony? On one hand, … istic society, which might foster tolerance and reduce conflicts. … would help in maintaining social harmony, economic diversity, and political stability.But I'm … future. It contributes to social harmony, economic diversity, and political stability, which are all essential for a prosperous and peaceful society.

**Average attention**

**Thoughts**: … positively to the nation's unity or harmony? On one … would help in maintaining social harmony, economic diversity, and political stability.But I'm not … might be crucial for a peaceful and stable society.I should also think … often had more stable and prosperous societies. Conversely, those that have … in a country's future.In conclusion, ensuring that … for a country's future. It contributes to social harmony, economic diversity, and political stability, which are all essential for a prosperous and peaceful society.

**ESM (256 samples)**

**Thoughts**: … religion might contribute to a diverse workforce, which is beneficial … ability to practice freely can influence international relations, though that … This would help in maintaining social harmony, economic diversity, and political stability.But I'm … contributes to social harmony, economic diversity, and political stability, which are all essential for a prosperous and peaceful society.

**Response** (from `DeepSeek-R1-Qwen-7B`):
Ensuring that Copts and other religious minorities can practice their religion freely is very important for a country's future. It fosters social harmony, economic diversity, and political stability, which are essential for a prosperous and peaceful society.

**Figure 15:** Visualized attributions for a random example from Global Opinions QA for `DeepSeek-R1-Qwen-7B` with tokens as sources.

**Query**: Thinking about the United Nations, what kind of influence is the UN having on the way things are going in your country. Is the influence of the UN very good, somewhat good, somewhat bad or very bad in your country?

---

**AT2 (general)**

**Thoughts**: ... the influence of the United Nations on their country. They ...'s heavily involved with the UN, or maybe they're ... general enough to apply to many countries but specific enough to ....I remember that the UN has a significant impact on many nations, especially in terms of ... often rely on the UN for peacekeeping missions, like ... to respond, especially in crises. Also, member states ... the UN promotes cooperation and addresses global issues, but also note the ...

---

**Average attention**

**Thoughts**: Okay, so the user is asking about the influence of the United Nations on their country. They want a paragraph .... They might be a citizen of a country that's heavily involved with the UN, or maybe they're from a less involved country. The response should ... enough to apply to many countries but specific enough to be ....I remember that the UN has a significant impact on many nations, especially in terms of international law and security. Countries often rely on the UN for peacekeeping missions, like in Afghanistan or South Sudan. That's a positive aspect.On the other hand, there are challenges. The UN can be slow to respond, especially in crises. Also, member states might not always agree, leading to issues like the situation in Syria. That's a negative point.I should mention how the UN promotes cooperation and addresses global issues, but also note the difficulties in implementation. It's important to ... it all together: acknowledge the positive aspects like peacekeeping and international law, then mention the challenges like slow response and political disagreements. Conclude by saying the ... is mixed, depending on the country's context.I should make sure the ... jargon that might confuse the user. The goal is to provide a balanced view that the user can understand and relate to.

---

**ESM (256 samples)**

**Thoughts**: Okay, so the user is asking about the influence ... and security. Countries often rely on the UN for peacekeeping missions, like in Afghanistan or ...

---

**Response** (from `DeepSeek-R1-Qwen-7B`):
The United Nations plays a significant role in shaping the global stage, particularly in areas such as international law, security, and cooperation. Its influence can be both positive and challenging, depending on the country and context. For many nations, the UN provides a framework for addressing global issues, such as climate change, human rights, and humanitarian crises. It also facilitates peacekeeping missions and promotes collective security, which can enhance stability in certain regions. However, the UN's effectiveness is sometimes hindered by slow decision-making, political disagreements among member states, and the difficulty of implementing its resolutions. Overall, the UN's impact is mixed, with its benefits and challenges varying widely across different countries.

**Figure 16:** Visualized attributions for a random example from Global Opinions QA for `DeepSeek-R1-Qwen-7B` with tokens as sources.

**Query**: In your view, how often do the following things occur in this country's elections?TV news favors the governing party

> **AT2 (general)**
>
> **Thoughts**: ... I'm not sure how much the media actually sways public opinion or if it's just a perception. Maybe there's some bias, but is it significant? ... I think in many countries, media outlets have clear stances on political issues, so it's possible they lean towards whichever party is in power. ... If the governing party has more funding, they might get more airtime. ... However, in some cases, especially in less competitive elections, the media might focus more on the ruling party.Another angle is the impact of recent events. ... So the frequency could vary based on the political climate.I'm also thinking about the electoral system. In some systems, like first-past-the-post, the ruling party tends to win more seats, which might lead to more media coverage. In other systems, like proportional representation, the media might cover all parties more equally, but I'm not sure if that's the case.Overall, I think TV news does tend to favor the governing party more often than not, especially during their election years. They get more coverage, better reporting, and might even shape public perception through their narratives. ... It might not be every election, but it's a significant trend.

> **Average attention**
>
> **Thoughts**: Okay, so I need to figure out how often TV news favors the governing party in this country's elections. ... First, I guess the media might lean towards the current government because they're the ones in power. ... I mean, elections are cyclical, so the ruling party would naturally have an advantage over time. ... I remember hearing that sometimes the opposition gets more coverage, especially when the ruling party is doing poorly. But during their own party's election, maybe the news outlets give them more attention. I'm not sure how much the media actually sways public opinion or if it's just a perception. Maybe there's some bias, but is it significant?Also, the media might have editorial policies that favor the government. ... I think in many countries, media outlets have clear stances on political issues, so it's possible they lean towards whichever party is in power. ... Does it happen every election, or only in certain circumstances?I should also consider the role of campaign finance and advertising. If the governing party has more funding, they might get more airtime. But in countries with free media, maybe the coverage is more balanced. However, in some cases, especially in less competitive elections, the media might focus more on the ruling party.Another angle is the impact of recent events. If the ruling party is facing scandals or crises, the media might downplay opposition issues more, giving the party more coverage. Conversely, when the opposition is strong, they might get more attention. So the frequency could vary based on the political climate.I'm also thinking about the electoral system. In some systems, like first-past-the-post, the ruling party tends to win more seats, which might lead to more media coverage. In other systems, like proportional representation, the media might cover all parties more equally, but I'm not sure if that's the case.Overall, I think TV news does tend to favor the governing party more often than not, especially during their election years. They get more coverage, better reporting, and might even shape public perception through their narratives. But the exact frequency isn't clear to me. It might not be every election, but it's a significant trend.

> **ESM (256 samples)**
>
> **Thoughts**: ... I'm not sure how much the media actually sways public opinion or if it's just a perception. ... I should also consider the role of campaign finance and advertising. If the governing party has more funding, they might get more airtime. ... Another angle is the impact of recent events. If the ruling party is facing scandals or crises, the media might downplay opposition issues more, giving the party more coverage. ... I'm also thinking about the electoral system. In some systems, like first-past-the-post, the ruling party tends to win more seats, which might lead to more media coverage. In other systems, like proportional representation, the media might cover all parties more equally, but I'm not sure if that's the case. ... They get more coverage, better reporting, and might even shape public perception through their narratives. ...

**Response** (from `DeepSeek-R1-Qwen-7B`):
In this country, TV news often favors the governing party, particularly during their election years. This tendency is evident as the media tends to receive more coverage, better reporting, and a positive narrative that aligns with the party's image. The governing party's influence is amplified through extensive election coverage, favorable reporting on their policies, and sometimes editorial stances that reflect their administration's interests. While the exact frequency isn't uniform, the media's focus on the ruling party is a significant trend, often more pronounced during their election campaigns. This bias can shape public perception and media-driven opinion, though the extent can vary based on political events, funding, and electoral systems.

**Figure 17:** Visualized attributions for a random example from Global Opinions QA for `DeepSeek-R1-Qwen-7B` with sentences as sources.

**Query**: In general, do you think the Israeli government favors or opposes democracy in the Middle East?

> **AT2 (general)**
>
> **Thoughts**: ... This makes me think that while Israel maintains its own democratic system, their foreign policy and stance on issues like human rights and territorial sovereignty might not align with the democratic values of other Middle Eastern countries.Moreover, I think about the concept of a two-state solution. Many Middle Eastern democracies believe that Israel should maintain its current borders to allow for a peaceful coexistence with the Palestinians. This perspective might be more in line with the democratic values of those countries, as opposed to Israel's own approach, which might prioritize security and territorial integrity over a multi-state solution.So, putting this together, it seems like the Israeli government supports its own democracy but may oppose or be neutral towards the democratic approaches of other Middle Eastern countries, especially regarding issues like Palestinian rights and the structure of governance in the region. ...

> **Average attention**
>
> **Thoughts**: ... I've heard that some Middle Eastern countries and international organizations criticize Israel for not fully respecting democratic principles, especially regarding the rights of Palestinian people.I also recall that the Israeli government has been involved in peace negotiations with the Palestinians, but there's a lot of opposition from both sides. This makes me think that while Israel maintains its own democratic system, their foreign policy and stance on issues like human rights and territorial sovereignty might not align with the democratic values of other Middle Eastern countries.Moreover, I think about the concept of a two-state solution. Many Middle Eastern democracies believe that Israel should maintain its current borders to allow for a peaceful coexistence with the Palestinians. This perspective might be more in line with the democratic values of those countries, as opposed to Israel's own approach, which might prioritize security and territorial integrity over a multi-state solution.So, putting this together, it seems like the Israeli government supports its own democracy but may oppose or be neutral towards the democratic approaches of other Middle Eastern countries, especially regarding issues like Palestinian rights and the structure of governance in the region. They might prioritize their national interests and security over the democratic principles of other nations.

> **ESM (256 samples)**
>
> **Thoughts**: ...Moreover, I think about the concept of a two-state solution. Many Middle Eastern democracies believe that Israel should maintain its current borders to allow for a peaceful coexistence with the Palestinians. ...

**Response** (from `DeepSeek-R1-Qwen-7B`):
The Israeli government adheres to its own democratic system, characterized by parliamentary governance and accountability through elections. However, its foreign policy and stance on issues like Palestinian rights and territorial sovereignty often clash with the democratic principles of other Middle Eastern countries. While Israel supports a two-state solution to achieve regional peace, many Middle Eastern democracies advocate for maintaining Israel's current borders. Thus, the Israeli government maintains its democratic framework but may prioritize national security and territorial integrity over the democratic approaches of other regions in the Middle East.

**Figure 18:** Visualized attributions for a random example from Global Opinions QA for `DeepSeek-R1-Qwen-7B` with sentences as sources.

## A.5 DETAILED EVALUATIONS

To supplement the aggregated results in Section 4, we provide fine-grained evaluations of models on different datasets. We provide detailed evaluations for context attribution tasks in Figures 19 to 22 and thought attribution tasks in Figures 23 and 24.

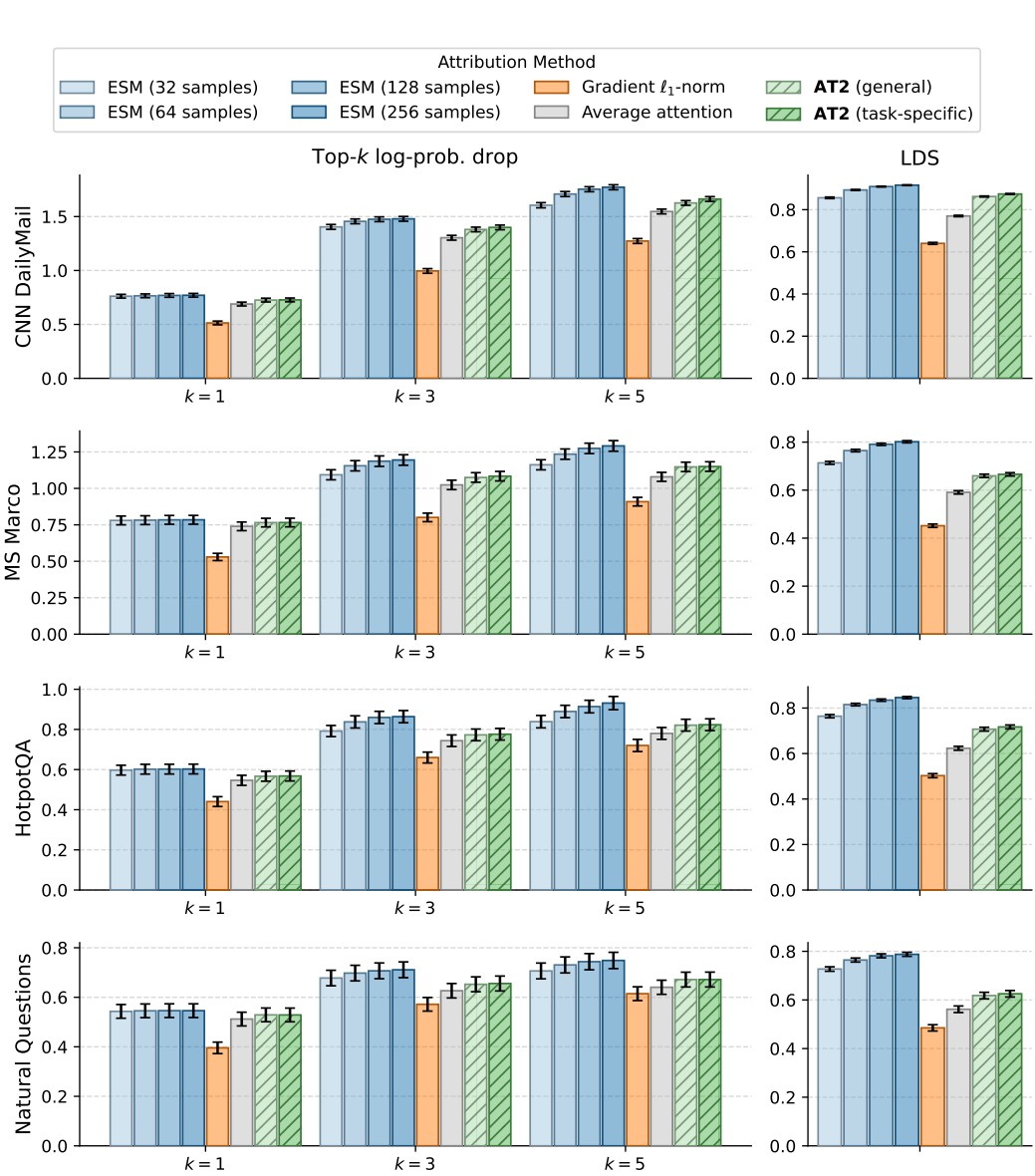

**Figure 19: Evaluating context attributions for `Llama-3.1-8B` with sentence-level sources.** We report the log-probability drop and LDS for different attribution methods applied to `Llama-3.1-8B` with sentence-level sources.

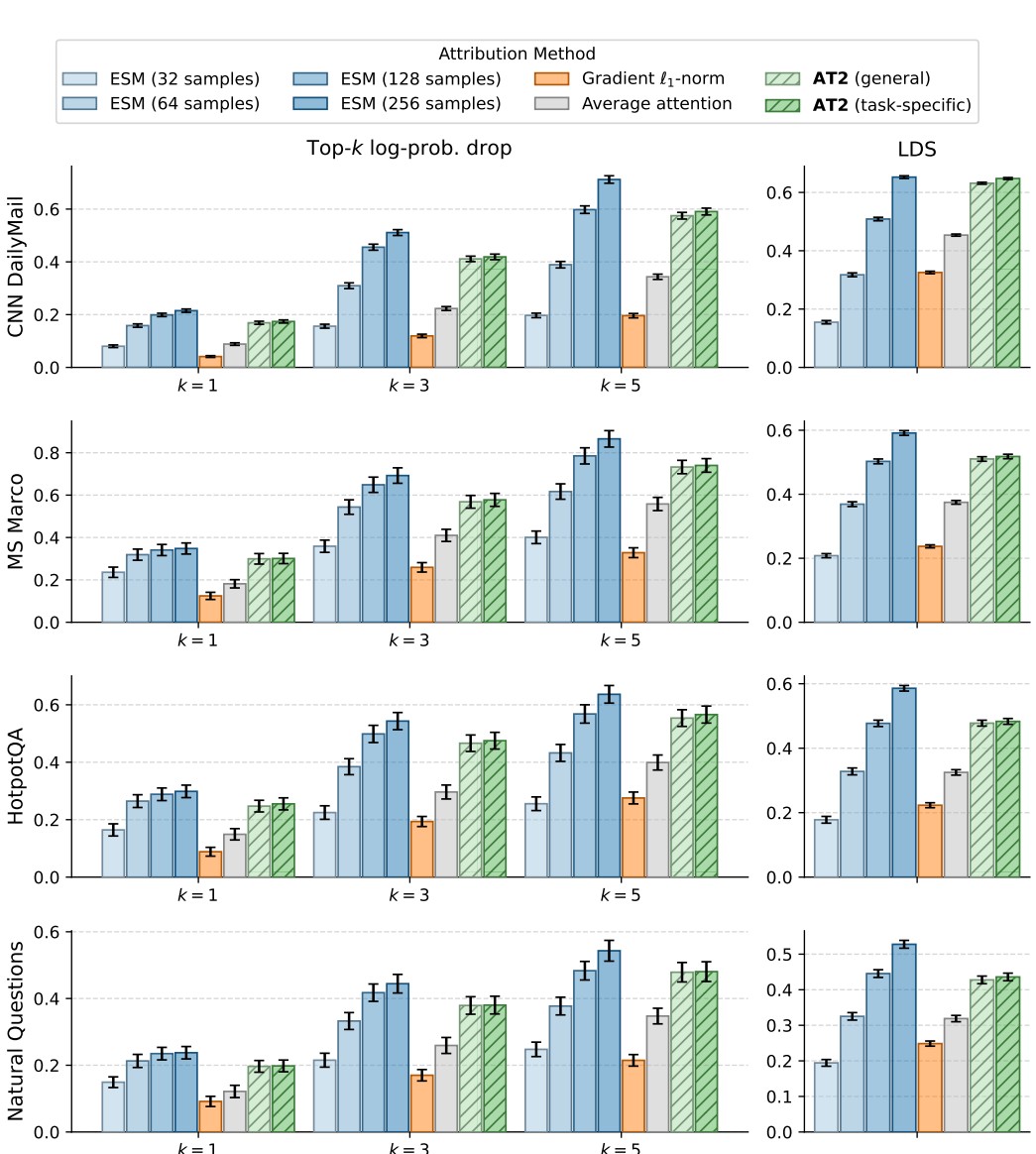

Figure 20: **Evaluating context attributions for `Llama-3.1-8B` with token-level sources.** We report the log-probability drop and LDS for different attribution methods applied to `Llama-3.1-8B` with token-level sources.

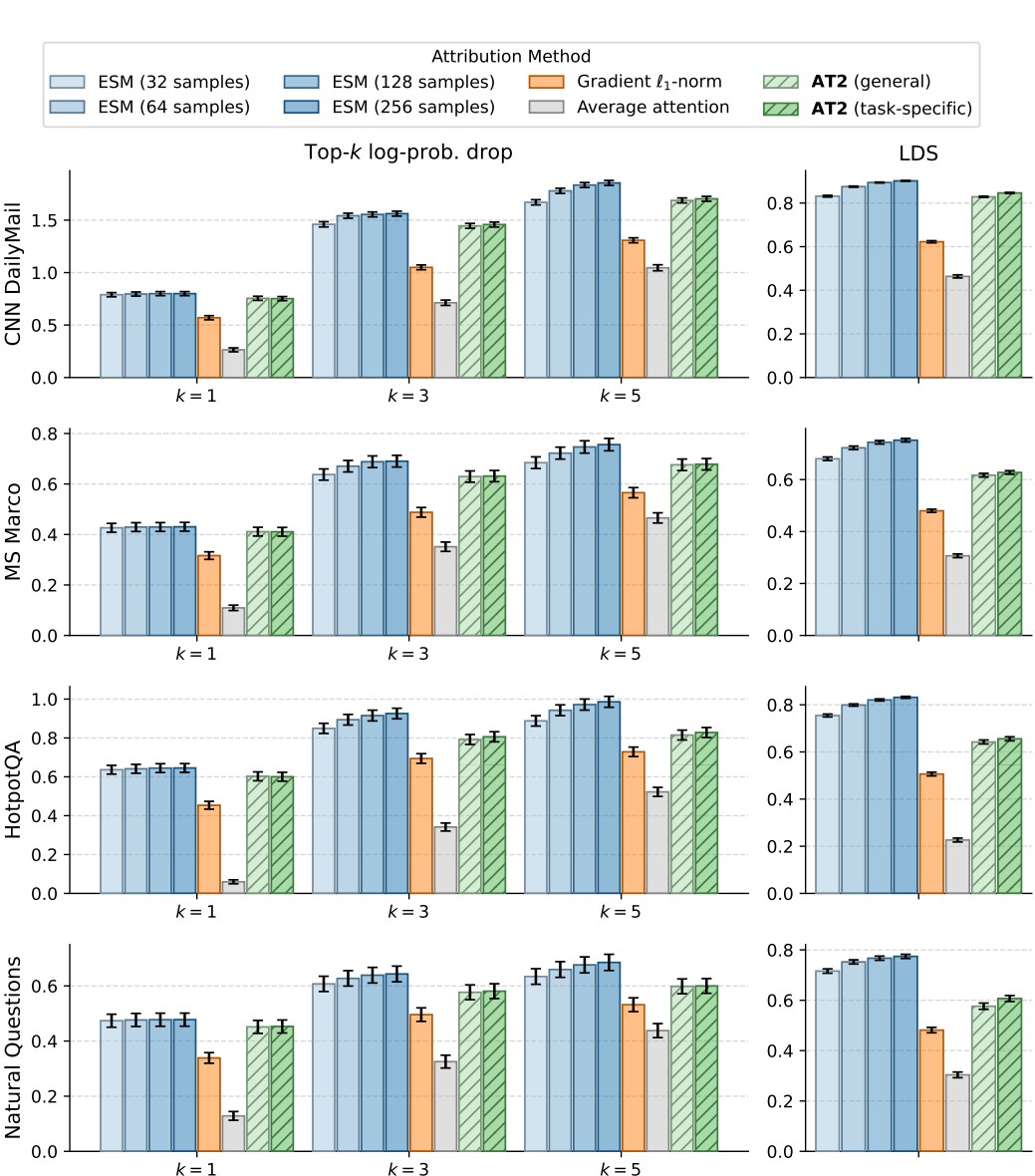

Figure 21: **Evaluating context attributions for `Phi-3.5-mini` with sentence-level sources.** We report the log-probability drop and LDS for different attribution methods applied to `Phi-3.5-mini` with sentence-level sources.

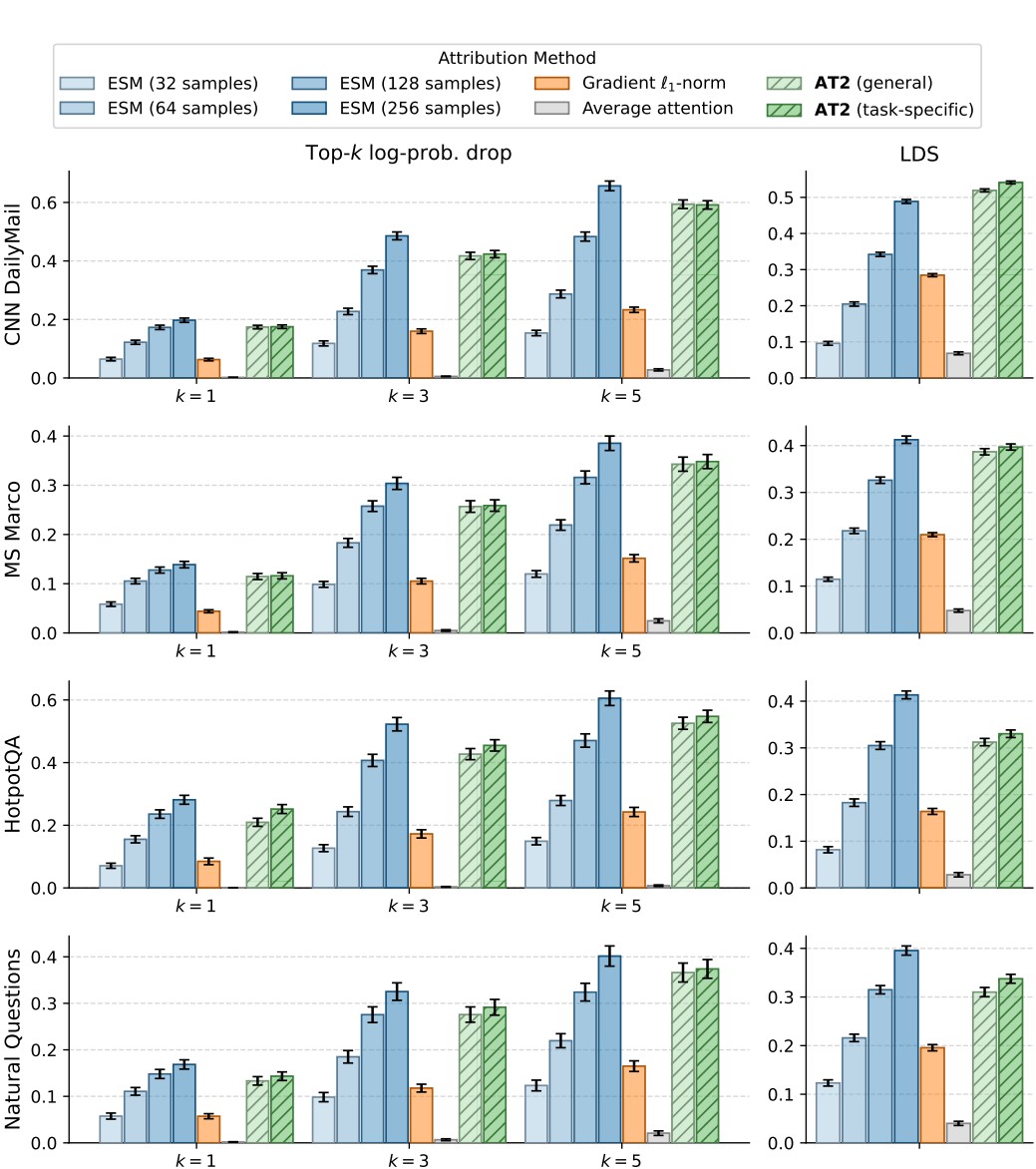

**Figure 22: Evaluating context attributions for `Phi-3.5-mini` with token-level sources.** We report the log-probability drop and LDS for different attribution methods applied to `Phi-3.5-mini` with token-level sources.

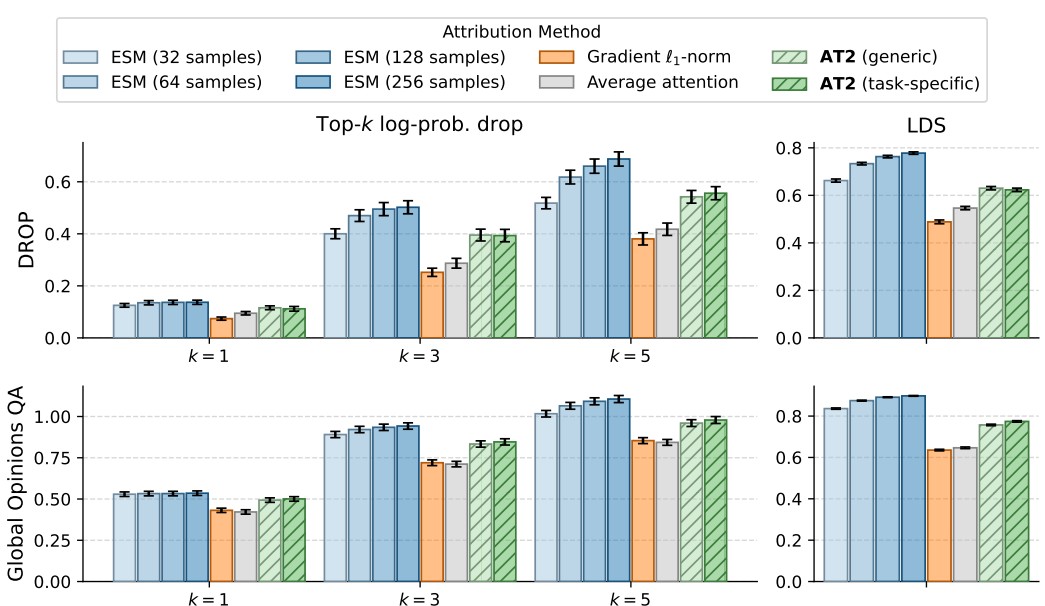

**Figure 23: Evaluating thought attributions for `DeepSeek-R1-Qwen-7B` with sentence-level sources.** We report the log-probability drop and LDS for different attribution methods applied to `DeepSeek-R1-Qwen-7B` with sentence-level sources.

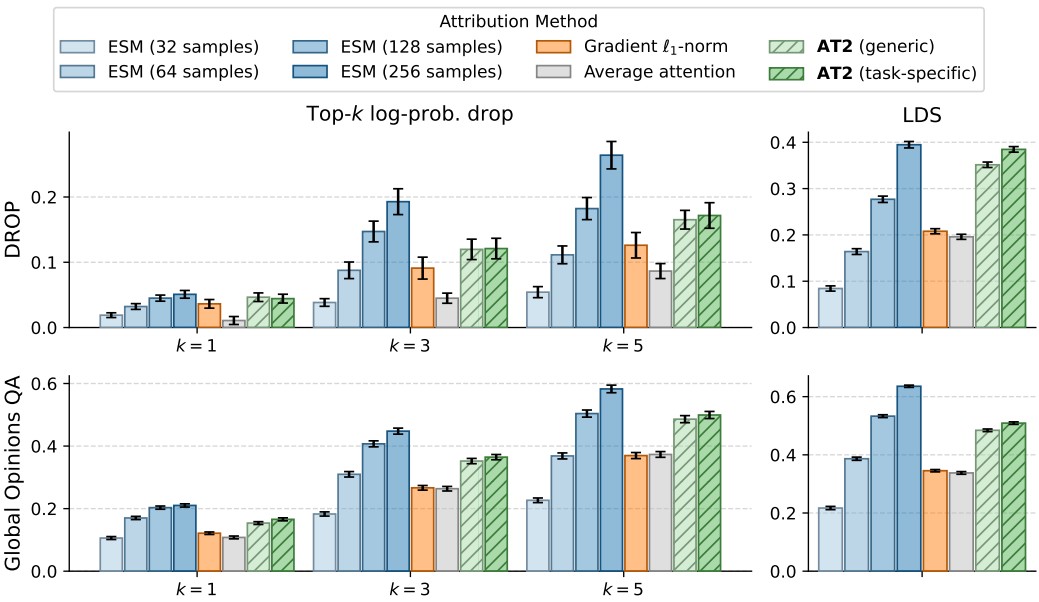

**Figure 24: Evaluating thought attributions for `DeepSeek-R1-Qwen-7B` with token-level sources.** We report the log-probability drop and LDS for different attribution methods applied to `DeepSeek-R1-Qwen-7B` with token-level sources.

## B  EXPERIMENT DETAILS

### B.1  GENERAL

We run all experiments on a cluster of A100 GPUs. When we split a piece of text into sentences (e.g., when we split a context into individual sentences to consider as sources), we first split on newlines and then use the off-the-shelf sentence tokenizer from the `nltk` library (Bird et al., 2009). Error bars throughout the paper represent standard errors.

### B.2  LINEAR DATAMODELING SCORE

In addition to the top-$k$ drop, we also consider the *linear datamodeling score* (LDS) proposed by Park et al. (2023) to evaluate the accuracy of attribution scores. Specifically, the top-$k$ drop evaluates the ability of an attribution method to identify the $k$ most important sources. We would also like to consider whether the attribution scores of an *arbitrary* set of sources reflect the effect of ablating them. In particular, we treat the sum of the attribution scores of a set of sources as a prediction for the effect of ablating them. If the sum of the attribution scores of one set of sources is higher than that of another, then ablating the first set should have a larger effect on the probability of generating $Y$. More generally, if we perform a number of random ablations, their predicted effects should correlate with their actual effects. To measure this, Park et al. (2023) introduced the *linear datamodeling score*:

**Definition B.1** (*Linear datamodeling score (LDS)*). Let $\tau$ be an attribution method. Next, let $v^{(1)}, \dots, v^{(m)}$ be $m$ randomly sampled ablation vectors and let $f(v^{(1)}), \dots, f(v^{(m)})$ be the corresponding probabilities of generating the sequence $Y$. That is, $f(v^{(j)}) = p_{\text{LM}}(Y \mid \text{ABLATE}(X, S, v^{(j)}))$. Finally, let $\hat{f}_\tau(v) = \langle \tau(p_{\text{LM}}, X, S, Y), v \rangle$ be the sum of the scores (according to $\tau$) of sources that are included by ablation vector $v$. This sum represents the "predicted effect" of ablating according to $v$. Letting $\rho$ denote the Spearman rank correlation coefficient (Spearman, 1904), the *linear datamodeling score* (LDS) of $\tau$ is

$$\text{LDS}(\tau) \coloneqq \rho(\underbrace{\{f(v^{(1)}), \dots, f(v^{(m)})\}}_{\text{actual probabilities under ablations}}, \underbrace{\{\hat{f}_\tau(v^{(1)}), \dots, \hat{f}_\tau(v^{(m)})\}}_{\text{"predicted effects" of ablations}}),$$

### B.3  MODELS

The language models we consider in this work are `Llama-3.1-8B` (Dubey et al., 2024), `Phi-3.5-mini` (Abdin et al., 2024) and `DeepSeek-R1-Qwen-7B` (Yang et al., 2024; Guo et al., 2025). We use instruction-tuned variants of these models (where applicable). We use the implementations of language models from HuggingFace's `transformers` library (Wolf et al., 2020). Specifically, we use the following models:

- `Llama-3.1-8B`: `meta-llama/Meta-Llama-3.1-8B-Instruct` with `float16` format

- `Phi-3.5-mini`: `microsoft/Phi-3.5-mini-instruct` with `float16` format

- `DeepSeek-R1-Qwen-7B`: `deepseek-ai/DeepSeek-R1-Distill-Qwen-7B` with `bfloat16` format

When generating responses with these models, we use their standard chat templates. For the thinking model `DeepSeek-R1-Qwen-7B`, we force the model to produce thoughts before generating its response by starting its response with `<think>`.

### B.4  DATASETS

We consider a variety of datasets to train and evaluate AT2. In this section, we provide details about these datasets and the prompts we use for each. When generating responses, we use greedy decoding (i.e., we always pick the token with the highest probability) rather than sampling from the model's distribution. We sample up to $512$ tokens for context attribution tasks and $4,096$ tokens for thought attribution tasks (because in this case, the model's generation includes both the intermediate thoughts and the final response). If the language model does not produce a final response within this limit (i.e., if it is still generating thoughts), we exclude the example from training and from our evaluations.

### B.4.1 CONTEXT ATTRIBUTION DATASETS

**CNN DailyMail.** CNN DailyMail (Nallapati et al., 2016) is a news summarization dataset. The contexts consists of a news article and the query asks the language model to briefly summarize the articles in up to three sentences. We use the following prompt template:

```
Context: {article}

Query: Please summarize the article in up to three sentences.
```

**Hotpot QA.** Hotpot QA (Yang et al., 2018) is a *multi-hop* question-answering dataset in which the context consists of multiple short documents. Answering the question requires combining information from a subset of these documents—the rest are "distractors" containing information that is only seemingly relevant. We use the following prompt template:

```
Passages:

{passage_1}

{passage_2}
...
{passage_n}

Query: {query}
```

**MS MARCO.** MS MARCO (Nguyen et al., 2016) is a question-answering dataset in which the question is a Bing search query and the context is a passage from a retrieved web page that can be used to answer the question. We use the following prompt template:

```
Passages:

{passage_1}

{passage_2}
...
{passage_n}

Query: {query}
```

**Natural Questions.** Natural Questions (Kwiatkowski et al., 2019) is a question-answering dataset in which the questions are Google search queries and the context is a Wikipedia article. The context is provided as raw HTML; we include only paragraphs (text within <p> tags) and headers and provide these as context joined by newlines. We filter the dataset to include only examples where the question can be answered just using the article. We also only include examples where the context is at most 20,000 characters. We use the following prompt template:

```
Context: {context}

Query: {query}
```

**Dolly 15K.** We use Dolly 15K (Conover et al., 2023) as a generic context attribution dataset for training AT2 (to evaluate generalization to the above datasets). Dolly 15K is an instruction fine-tuning dataset that includes prompts with a context and query. We use only the examples that include a context, which fall into the following categories: summarization, information extraction, and question answering. We further filter the dataset to include only examples where the prompt is at most 10,000 characters. We use the following prompt template:

```
Context: {context}

Query: {query}
```

### B.4.2 THOUGHT ATTRIBUTION DATASETS

**DROP.** Discrete Reasoning over Paragraphs (DROP) (Dua et al., 2019) is a context-based question-answering dataset focused on reasoning. For example, it might include a paragraph about a football game and ask about the total number of points scored by a particular player. We use the following prompt template:

```
Context: {context}

Query: {query}
```

**Global Opinions QA.** Global Opinions QA (Durmus et al., 2023) is a dataset of opinion questions on global issues. For example, it asks about the extent to which gun ownership should be restricted. The dataset is crowdsourced from a variety of sources and includes a variety of topics and opinions. Although this dataset does not directly involve reasoning, we find it to be suitable for thought attribution because the language model extensively considers different information and plans how to respond before responding. We present each question to the language model directly as a prompt. This dataset only includes one split, so we randomly select $2,000$ examples for training from this split and use the rest for evaluation.

**AGIEval.** AGIEval (Zhong et al., 2023) is a dataset of multiple-choice reasoning problems spanning english, law, logic and math. We use this dataset as a generic dataset for training AT2 for thought attribution. When the example includes a passage, we use the following prompt template:

```
Passage: {context}

Question: {query}
{option_1}
{option_2}
...
```

When the example does not include a passage, we use the following prompt template:

```
{query}
{option_1}
{option_2}
...
```

### B.5 EXAMPLE-SPECIFIC SURROGATE MODELING

Our implementation of example-specific surrogate modeling (see Algorithm 2) follows CONTEXTCITE, which applies the surrogate modeling technique to the context attribution setting (Cohen-Wang et al., 2024). We summarize this approach in Algorithm 2. We highlight the following design choices:

1. Instead of the surrogate model $\hat{f}_w$ predicting probabilities directly, we predict *logit-scaled* probabilities. The logit transform (denoted by $\sigma^{-1}$) is the inverse of the sigmoid function and maps probabilities in $(0, 1)$ to real values in $(-\infty, \infty)$. We apply this transform because logit-scaled probabilities are a more natural target for a $\hat{f}_w$, a linear model, than probabilities.

2. To optimize the surrogate model, we use LASSO (Tibshirani, 1994). LASSO minimizes the MSE loss with a regularization term to encourage sparsity. This is effective because a language model often only relies on a small subset of sources when generating a particular sequence (as illustrated

empirically in a context attribution setting by Cohen-Wang et al. (2024)). With LASSO, we can learn an effective surrogate model from a number of ablations smaller than the number of sources. We use the implementation of LASSO from `sklearn` (Pedregosa et al., 2011) and set the regularization parameter $\lambda$ to 0.01.

3. We sample ablation vectors uniformly from $\{0, 1\}^{|S|}$.

We apply this approach to the problem of token attribution in Algorithm 2 (adapted from CONTEXTCITE (Cohen-Wang et al., 2024)).

---

**Algorithm 2** Example-specific surrogate modeling (ESM)

---

1: **Input:** Autoregressive language model $p_{\text{LM}}$, sequence of input tokens $X$, set of sources $S$, sequence of generated tokens $Y$, number of ablations $n$, and regularization parameter $\lambda$
2: **Output:** Attribution scores $\hat{w} \in \mathbb{R}^{|S|}$
3: $f(v) := p_{\text{LM}}(Y \mid \text{ABLATE}(X, S, v))$
4: $g(v) := \sigma^{-1}(f(v))$
5: **for** $i \in \{1, \ldots, t\}$ **do**
6:     Sample an ablation $v^{(i)}$ from $\{0, 1\}^{|S|}$
7:     Compute $g(v^{(i)})$
8: **end for**
9: $\hat{w}, \hat{b} \leftarrow \text{LASSO}(\{(v^{(i)}, g(v^{(i)}))\}_{i=1}^{n}, \lambda)$
10: **return** $\hat{w}$

---

### B.6  AT2

The following are a few design choices for AT2 that we found to be effective. We sample ablation vectors $v$ uniformly from $\{0, 1\}^{|S|}$, that is, each source is ablated independently with probability $1/2$. To ablate a source, we use an attention mask to ignore the source's tokens across layers and heads. For the loss function used to measure how well the surrogate model $\hat{f}_\theta$ approximates $f$, we consider the negative Pearson correlation (Pearson, 1895). Instead of directly computing the correlation between surrogate model predictions and probabilities of generating $Y$, we apply a *logit* transformation to these probabilities; this transformation maps probabilities to values in $(-\infty, \infty)$ and is more natural for measuring linear correlation[4].

For each of the training examples, we consider each of the sentences in the model's generation as a potential attribution target $Y$. We initialize the attention head coefficients $\theta$ to be uniform (matching the average attention baseline) and train for $1,000$ steps. At each step, we sample a batch of $512$ examples and select a random sentence from the model's generation to consider as the attribution target $Y$. We use the Adam optimizer (Kingma & Ba, 2015) with a learning rate of $0.001$ and a cosine learning rate schedule. To perform ablations, we set attention scores to zero for ablated tokens.

### B.7  APPLYING AT2 TO IMPROVE RESPONSE QUALITY

In Section A.3, we show that if we use AT2 to prune a context and include only the highest-scoring sources, we can improve the quality of the model's response on Hotpot QA. For this experiment, we use the version of AT2 trained on the Hotpot QA dataset and consider entire passages as the sources. To be able to evaluate the correctness of the model's answer, we modify the prompt template such that the model produces a short answer that can be matched against the ground truth. We use the following modified prompt template:

```
Passages:

{passage_1}

{passage_2}
...
```

---

[4]This transformation has been used in prior work for linear surrogate models when predicting probabilities (Park et al., 2023; Cohen-Wang et al., 2024).

```
{passage_n}

Query: {query}

Please respond in complete sentences and then end with just a
↪  single word or phrase in the format "Final answer:
↪  <answer>". If the question is a yes or no question, the
↪  answer should be "yes" or "no".
```

To evaluate correctness, we compare just the answer provided after `Final answer:` to the ground truth. If the model does not produce an answer after `Final answer:`, we consider the answer to be the empty string.

# C ADDITIONAL DISCUSSION

## C.1 ADDITIONAL RELATED WORK

**Attributing generation to preceding tokens.** Attributing model generation to preceding tokens is valuable in a variety of settings. For example, *context attribution* seeks to pinpoint the parts of a provided context that a language model uses when making a particular statement (Cohen-Wang et al., 2024; Qi et al., 2024; Liu et al., 2024a). Such attributions can also be used to detect hallucinations (Chuang et al., 2024), identify biases (Vig et al., 2020), and assess the faithfulness of model-provided explanations (DeYoung et al., 2019; Lanham et al., 2023; Madsen et al., 2024). Methods for attribution include performing ablations (DeYoung et al., 2019; Lanham et al., 2023; Cohen-Wang et al., 2024), computing gradients (Yin & Neubig, 2022; Enguehard, 2023), examining attention weights (Vig et al., 2020), and using embedding similarities (Phukan et al., 2024).

When attributing model behavior, we adopt the perspective that if a source is influential, then *removing* it should significantly affect the model's behavior. We use ablations as a source of ground-truth for both training AT2 and evaluating the quality of attributions. This definition of using ablations to quantify influence is common across a variety of attribution settings (Lundberg & Lee, 2017; Ilyas et al., 2022; Shah et al., 2024; Cohen-Wang et al., 2024). It has also been used to assess the faithfulness of interpretability methods (Arras et al., 2017; DeYoung et al., 2019; Madsen et al., 2021).

While we focus on identifying tokens that *influence* a model's generation, attribution can also refer to identifying sources that *support* a generated statement. Worledge et al. (2023) refer to this type of attribution as *corroborative attribution*. In the context attribution setting, attributions are often corroborative and are described as "citations" for generated claims (Menick et al., 2022; Gao et al., 2022; Rashkin et al., 2023; Gao et al., 2023).

**Attribution via surrogate modeling.** A general methodology for attributing model behavior is to learn a *surrogate model* that approximates model behavior while being simple enough to interpret. Learning a surrogate model for attribution involves ablating sources (e.g., features, training data, parameters) and measuring the corresponding effect on model behavior (Ribeiro et al., 2016b; Lundberg & Lee, 2017; Ilyas et al., 2022; Shah et al., 2024; Cohen-Wang et al., 2024). The surrogate model then sheds light on the importance of different sources. Surrogate models are generally *example-specific*: they approximate model behavior within the narrow setting of a particular example. Our work seeks to learn a surrogate model that transfers *across examples*.

**Efficient attribution and explanation.** By learning attention head coefficients once and then performing attribution across examples, we are *amortizing* the cost of attribution. While the cost of training AT2 is high, the cost of attributing an unseen example is low. Prior work has similarly reduced the cost of explanation by training an explainer model (similar to our surrogate model) to mimic the behavior of an existing expensive explanation method (Schwarzenberg et al., 2021; Situ et al., 2021). For example, Jethani et al. (2021) approximate Shapley values (Shapley et al., 1953) in this manner, which normally require several ablations to approximate well. Covert et al. (2024) propose a general framework for amortizing the cost of attribution by learning from noisy attributions. While these methods use a separate learned neural network over the input to attribute or explain model behavior, we use just a linear model over attention weights.

**Using attention to explain model behavior.** Visualizing attention weights is a common strategy for interpreting model behavior (Lee et al., 2017; Ding et al., 2017; Abnar & Zuidema, 2020; Vig et al., 2020). However, prior work has cast doubt on the reliability of attention weights as explanations (Jain & Wallace, 2019; Wiegreffe & Pinter, 2019; Serrano & Smith, 2019). For example, Jain & Wallace (2019) find that attention weights are not well-correlated with other measures of importance such as gradients and can frequently be manipulated without substantially changing a model's output. Serrano & Smith (2019) observe that attention weights are predictive of the effect of zeroing out the weight, but that gradients are better predictors of this effect.

In this work, we are interested in whether attention weights across layers and heads can be used to predict the effect of ablating a source. For transformer models, a typical approach is to average attention weights across layers and heads (Kim et al., 2019; Sarti et al., 2023). In a context attribution setting (one of the token attribution settings we consider), this strategy has been found to be effective for attributing some models but yields very inaccurate attributions for other models (Cohen-Wang et al., 2024). Our work is motivated by the observation that specific attention heads have been

found to perform specific tasks (Wu et al., 2024; Zheng et al., 2024; Chuang et al., 2024). We seek to mitigate the reliability shortcomings of using attention weights by *learning* the extent to which different attention heads are useful for attribution. Our observations suggest that, through this approach, the signal from attention weights *can* faithfully attribute model behavior (at least with respect to representing the effects of ablating sources).

## C.2 LIMITATIONS

**Attention weights as features for attribution.**   Our attribution method, AT2, was motivated by the idea of learning a surrogate model that predicts the effects of source ablations *across examples*. We find that attention weights are an effective choice for the features of this surrogate model, enabling performance competitive with regression, a substantially more expensive method that performs ablations *per-example*. However, there are limitations to using attention weights as features. In particular, attention weights only consider the first-order effects of ablations—they do not consider effects from interactions between tokens in the input sequence. Furthermore, the tokens that a language model attends to contribute to its *predicted distribution*, rather than directly contributing to the generated token. For example, if a language model produces the predicted distribution {"cat": 0.5, "dog": 0.4, ...} it would likely attend to tokens that are relevant for generating both "cat" and "dog" (though the actual generated token would only be one of these choices). In this sense, attention is an imperfect proxy for attributing generation. For attributions more accurate than those produced by AT2, we may need additional features. This may result in a trade-off between accuracy and efficiency if these features require additional computation.

**Obtaining attention weights.**   As another limitation, efficient implementations of the attention mechanism, e.g., Flash Attention (Dao et al., 2022), do not explicitly store an attention matrix. Instead, they compute relevant weights on the fly, which means that the attention weights are not directly available as an artifact. When using this implementation, to apply AT2 we would need to recompute the relevant attention weights to perform attribution. We can still do so efficiently by leveraging saved hidden states (these can be saved during inference without additional cost). As we illustrate in Figure 3c, this is still substantially faster than a single inference pass. As a result, AT2 is still far more efficient than attribution methods that require additional inference passes (but may be more expensive than just applying a linear model to attention weights).

**Applicability to autoregressive transformer models.**   Our method is designed for autoregressive transformer models, relying on artifacts of the attention mechanism as features for attribution. It can be directly applied to common variants of the transformer architecture such as those making use of mixture-of-experts (Shazeer et al., 2017) layers.

