# OpenReview forum: "Learning to Attribute with Attention"
_ICLR.cc/2026/Conference — ICLR 2026 Conference Withdrawn Submission_

### Official Review · Reviewer_Y4X7 · 2025-10-28

**Soundness:** 2
**Presentation:** 2
**Contribution:** 2
**Rating:** 2
**Confidence:** 5

**Summary:**

AT2 learns per-head coefficients over attention weights to predict how ablating prior tokens would change a model’s output, then uses those weights for fast attribution on new examples. Empirically, it matches example-specific ablation surrogates while beating gradient and average-attention baselines, and is far more efficient at inference time

**Strengths:**

* Simple, generalizable signal: Treats attention heads as features and learns how much to trust each head, aligning with evidence that heads differ functionally.

* Strong efficiency: AT2 is >2× faster than one forward pass and ~8× faster than one backward pass; concrete timings are reported on Llama-3.1-8B/HotpotQA.

* Competitive accuracy: Performs comparably to example-specific surrogate modeling and better than gradient/average-attention across context and thought attribution.

**Weaknesses:**

* Although per-head learning helps, attention remains a debated proxy for explanation. Also AT2 assumes the effect of ablating sources is a linear, additive function and then learns a single, input-independent set of head coefficients to predict it from attention weights. This collapses inherently non-linear, context-dependent interactions between tokens and heads into fixed, first-order features.

* Lacks thorough comparisons to contemporary attribution method like Oren Barkan, Yonatan Toib, Yehonatan Elisha, Jonathan Weill, and Noam Koenigstein. Llm ex- plainability via attributive masking learning. In Findings of the Association for Computational Linguistics: EMNLP 2024, pp. 9522–9537, 2024. Please see the recent works on attribution methods and compare with them how your method performs.

* How AT2 behaves when applied to larger models above 10B and how the computation time varies.

**Questions:**

same as above

---

### Official Review · Reviewer_FLah · 2025-10-28

**Soundness:** 3
**Presentation:** 3
**Contribution:** 3
**Rating:** 6
**Confidence:** 4

**Summary:**

This paper introduces AT2 (Learning to Attribute with Attention), a method for efficient and faithful token-level attribution in large language models. AT2 is motivated by the observation that attention heads vary in their usefulness for attribution—some correlate closely with causal influence, while others do not.

The method trains a global surrogate that learns head-specific coefficients to linearly weight attention distributions across layers, approximating the log-probability change induced by token ablations. Once trained, the surrogate provides attribution for new examples without performing additional ablations or gradient computations, achieving near-ablation faithfulness while being significantly faster. Experiments on context and thought attribution tasks using Phi-3.5-Mini, Llama-3.1-8B, and DeepSeek-R1-Qwen-7B demonstrate that AT2 generalizes across models and tasks.

**Strengths:**

1. **Clear motivation and conceptual framing:** The paper clearly identifies that not all attention heads contribute equally to causal influence, and formulates attribution as learning to weight heads by their usefulness.

2. **Practical efficiency:** AT2 amortizes ablation-based attribution, providing faithful approximations with significant speedup.

3. **Faithful yet simple formulation:** The linear surrogate is interpretable, easy to train, and applicable to existing transformer architectures without modification.

**Weaknesses:**

1. **Head redundancy and insufficient analysis:** Many attention heads are highly correlated, yet the paper provides no analysis of how this redundancy affects the surrogate’s performance.

2. **Lack of surrogate architecture analysis:** The paper demonstrates that the linear surrogate works but does not justify _why_ this specific structure is optimal compared to non-linear or hierarchical alternatives.

3. **Architecture sensitivity untested:** AT2’s features depend directly on the target model’s attention topology (number of heads, layer depth, sparsity). The paper does not study whether the method generalizes across models with distinct attention configurations.

**Questions:**

**Questions for the Authors**

1.  How stable are the learned coefficients across random seeds or training subsets?  Do the same heads consistently emerge as useful for attribution?

2. Have you considered comparing AT2 with attribution methods that use gradient-weighted attention features? It would be valuable to include comparisons with Attention × Grad variants such as Att×Att-Grad and Grad-SAM [1].

3. How sensitive is AT2’s performance to correlated or redundant attention heads?

4. How does AT2 behave under models with different attention architectures (e.g., sparse, grouped, or multi-query attention)?

[1] Oren Barkan et al., Grad-sam: Explaining transformers via gradient self-attention maps, In CIKM, 2021

**Additional Suggestions**

1. The notation in Algorithm 1 could be clarified and made more consistent with the equations in the main text. This would improve readability and alignment between the pseudo-code and the mathematical formulation.

---

### Official Review · Reviewer_h1vT · 2025-10-29

**Soundness:** 2
**Presentation:** 2
**Contribution:** 2
**Rating:** 2
**Confidence:** 4

**Summary:**

AT2 first learns how much to trust each attention head, treating the heads like features, and fits a simple linear predictor that estimates how the model’s output would change if you removed earlier tokens. After this one-time training, it reuses the same learned weights to produce fast attributions on new inputs. In experiments, it performs about as well as per-example ablation surrogates and better than gradient or average-attention baselines, while being very efficient: it takes less than half the time of a single forward pass and roughly one-eighth the time of a backward pass on Llama-3.1-8B with HotpotQA.

**Strengths:**

Treats attention heads as features and learns per-head trust; aligns with evidence that heads differ functionally.

Strong efficiency. AT2 is >2× faster than one forward and ~8× faster than one backward pass on Llama-3.1-8B/HotpotQA (timings reported).

Competitive accuracy. Comparable to example-specific surrogate modeling (ESM) and better than gradient and average-attention across context and thought attribution.

**Weaknesses:**

* Attention as explanation is debated. Prior work questions attention’s reliability, so per-head learning helps but doesn’t settle the proxy issue.

* AT2 assumes additive effects of source ablations and learns a single, input-independent matrix θ; this can miss non-linear, context-dependent interactions between tokens and heads.

* Empirical comparisons cover ESM, gradient, and average attention but several recent attribution methods are not included as baselines.

* Experiments target ≤8B models but comparing the behavior and timings for larger models will increase the reliability of the method.

**Questions:**

Please see the weakness as questions

---

### Official Review · Reviewer_AaoN · 2025-10-30

**Soundness:** 3
**Presentation:** 3
**Contribution:** 1
**Rating:** 2
**Confidence:** 3

**Summary:**

The paper proposes AT2, an amortized attribution method that learns global per-head attention weights using a small set of ablation-supervised examples, then applies these weights at test time to score source tokens/sentences. It evaluates on context attribution and “thought attribution,” comparing to example-specific surrogates, attention averaging, and gradient norms.

**Strengths:**

- The problem setup is very clear, the paper is well written.
- Amortizing per-example surrogates with shared parameters is practical.
- Has empirical gains over naïve baseline and has much lower test-time cost.

**Weaknesses:**

- Train a higher-level (amortized) surrogate to mimic ablation effects using attention-head features feels like a straightforward application of standard amortization/linear-surrogate ideas. The trade-off (speed vs. small fidelity drop) is unsurprising and largely incremental.
- A method can be simple, but then there should be more insight on why and how it works. At the moment, the paper reads as an engineering tweak rather than yielding interpretability insights.
- Results mostly compare a few operating points, it’s hard to see a compelling “dominant region” where AT2 is the obvious choice. Maybe show some kind of compute-normalized Pareto, e.g., plotting performance vs. test-time FLOPs/latency across baselines.

**Questions:**

- When AT2 fails vs. ESM, maybe analyze why and what happened (head misweighting? insufficient feature expressivity?).
- Is it possible to plot some Pareto curve across methods, maybe at some fixed test-time budgets?

---

### Official Review · Reviewer_thPH · 2025-11-01

**Soundness:** 3
**Presentation:** 3
**Contribution:** 3
**Rating:** 6
**Confidence:** 3

**Summary:**

The paper proposes a new token attribution method AT2. It assigns a learnable coefficient to each attention head, and the attribution score for an input token is the sum of attention scores from different attention heads weighted by their coefficient. The method is evaluated on context attribution and thought attribution. The results show that AT2 outperforms other methods.

**Strengths:**

The method is well-motivated and clearly presented.
While the training is expensive, the cost of the attribution at inference time is negligible.
The experimental results show that the method is effective and efficient.

**Weaknesses:**

The major problem I am concerned about is the rationalisation of measuring the attribution/influence 'ground truth' by ablating the input: 1) the corrupt sequence may change the semantic meaning and result in different attention behaviours, and 2) this method is good at seeing what one token does on its own, but it may struggle with teamwork, e.g., he influence of one token might be entirely dependent on the presence of another.

**Questions:**

It would be easy to train different non-linear models after collecting the datasets of full vs ablated attention scores. Do you try different model designs, though the linear model has the best efficiency

---

### Note · Authors · 2025-11-15

I have read and agree with the venue's withdrawal policy on behalf of myself and my co-authors.